# Clustering Items through Bandit Feedback
# Finding the Right Feature out of Many

**Maximilian Graf** [* 1]   **Victor Thuot** [* 2]   **Nicolas Verzelen** [2]

## Abstract

We study the problem of clustering a set of items based on bandit feedback. Each of the $n$ items is characterized by a feature vector, with a possibly large dimension $d$. The items are partitioned into two unknown groups, such that items within the same group share the same feature vector. We consider a sequential and adaptive setting in which, at each round, the learner selects one item and one feature, then observes a noisy evaluation of the item's feature. The learner's objective is to recover the correct partition of the items, while keeping the number of observations as small as possible. We provide an algorithm which relies on finding a relevant feature for the clustering task, leveraging the Sequential Halving algorithm. With probability at least $1 - \delta$, we obtain an accurate recovery of the partition and derive an upper bound on the budget required. Furthermore, we obtain an instance-dependent lower bound, which is tight in some relevant cases.

## 1. Introduction

We consider a sequential and adaptive pure exploration problem, in which a learner aims to cluster a set of items, each represented by a feature vector in $\mathbb{R}^d$. The items are partitioned into two unknown groups such that items within the same group share the same feature vector. The learner sequentially selects an item and a feature, and then observes a noisy evaluation of the chosen feature of that item. Given a prescribed probability $\delta$, the learner's objective is to collect enough information to recover the partition of the items with a probability of error at most $\delta$.

---

[*]Equal contribution   [1]Institut für Mathematik, Universität Potsdam, Potsdam, Germany [2]INRAE, Mistea, Institut Agro, Univ Montpellier, Montpellier, France. Correspondence to: Maximilian Graf <graf9@uni-potsdam.de>, Victor Thuot <victor.thuot@inrae.fr>.

*Proceedings of the 42$^{nd}$ International Conference on Machine Learning*, Vancouver, Canada. PMLR 267, 2025. Copyright 2025 by the author(s).

This problem arises in crowdsourcing platforms, where complex labeling tasks are decomposed into simpler sub-tasks, typically involving answering specific questions about an item – see (Ariu et al., 2024). A motivating example is image labeling: a platform sequentially presents an image to a user along with a simple question such as "Is this a vehicle?" or "How many wheels can you see?". The learner leverages these answers to classify the images into categories. In this setting, the images correspond to items that must be clustered, while questions correspond to features. This problem is a special case of the model studied in (Ariu et al., 2024), where the authors numerically demonstrate the advantage of an adaptive sampling scheme over non-adaptive ones. However, they do not establish the theoretical validity of their adaptive procedure.

From a theoretical perspective, our problem consists of clustering $n$ items based on sequential and adaptive queries to some of their $d$ features. Intuitively, the difficulty of the clustering task is driven by the differences between items in different groups across these $d$ features. In particular, it depends both on the magnitude of these differences and on their sparsity–that is, the number of features on which the items differ significantly.

In this work, we precisely characterize the sample complexity of the two-group[1] clustering task, in a fully adaptive setting. Our **main contributions** are as follows:

- We introduce the `BanditClustering` procedure – Algorithm 4. On the one hand, it outputs the correct partition of the items with a prescribed probability $1-\delta$. On the other hand, it adapts to the unknown means of the groups in order to sample at most the informative features. In Theorem 3.1, we provide a tight, non-asymptotic upper bound on its sample complexity as a function of $n$, $d$, $\log(1/\delta)$, and the difference between the means.

- Conversely, we establish in Section 4 an information-theoretic lower bound on the budget, which entails the optimality of `BanditClustering`.

From a high-level perspective, our algorithm operates in three steps: first, it identifies a pair of representative items

---

[1]In the core manuscript, we focus on $K = 2$ groups. We discuss how to extend our ideas to $K > 2$ clusters in Appendix C.

belonging to different groups; second, it selects a feature that best discriminates between the groups; and finally, it leverages this discriminative feature to cluster all items.

**Connection to good arm identification and adaptive sensing literature**. One of the key challenges is to achieve the trade-off between the budget used for identifying a good discriminative feature, and the budget used for the clustering task. We borrow techniques from the best-arm identification literature, specifically employing the Sequential Halving algorithm (Karnin et al., 2013) as a subroutine, leveraging its strong performance in settings where multiple arms are nearly optimal (Zhao et al., 2023; Katz-Samuels & Jamieson, 2020; Chaudhuri & Kalyanakrishnan, 2019). The first identification step — finding two items belonging to distinct groups — is closely related to the adaptive sensing strategies for signal detection, as studied in (Castro, 2014), where the problem is framed as a sequential and adaptive hypothesis testing task. Furthermore, our approach incorporates ideas from (Castro, 2014; Saad et al., 2023) to efficiently identify the most informative features for clustering.

**Connection to dueling bandits literature**. Our bandit clustering problem is an instance of a pure bandit exploration problem, where one can sample interaction between items and features. In that respect, it is also related to ranking (Saad et al., 2023) and dueling bandits in the online literature (Ailon et al., 2014; Chen et al., 2020; Heckel et al., 2019; Jamieson & Nowak, 2011; Jamieson et al., 2015; Urvoy et al., 2013; Yue et al., 2012; Haddenhorst et al., 2021) where the goal is to recover a partition of the items based on noisy pairwise comparisons. Some of the ranking procedures are based on estimating the Borda count (Heckel et al., 2019), some other procedures, such as (Saad et al., 2023), aim at adapting to the unknown form of the comparison matrix to reduce the total budget. In essence, our approach is related, as it seeks to balance the trade-off between identifying relevant entries and exploiting them for efficient comparisons.

**Connection to other bandit clustering problems**. Recent works (Yang et al., 2024; Thuot et al., 2025; Yavas et al., 2025) have investigated clustering in a bandit setting, where items must be clustered based on noisy evaluations of their feature vectors. However, in these settings, the entire feature vector of the chosen item is observed at each sampling step, whereas in our framework, only a single feature of a given item is observed per step. Our observation scheme enables a more efficient allocation of the budget by focusing on the most relevant features — those that best discriminate between groups. The trade-off between exploring relevant features and exploiting them for classification is at the core of our work. This allows us to cluster the items with a much lower observation budget than in (Yang et al., 2024; Thuot et al., 2025) — see the discussion section. Other authors

have previously introduced adaptive clustering problems for crowdsourcing (Ho et al., 2013; Gomes et al., 2011), although their settings does not directly relate to ours.

**Connection to online clustering of bandits**. We point out another line of works (Gentile et al., 2014; Li et al., 2019; Liu et al., 2022; Li et al., 2025) on the so-called *online clustering of bandits* problem – an instance of contextual linear bandit. This problem bears some resemblance to our setting, as it involves exploring a bandit environment with an underlying clustering structure among items. Still, there are two major differences with our problem: (1) the learner has no control over which items are presented at each time step, and (2) the algorithms (such as the CLUB Algorithm and its extensions) are designed and evaluated in a cumulative regret setting.

**Organization of the manuscript**. The model is introduced along with notation in the following Section 2. In Section 3, we describe our procedure and analyze its sample complexity. Section 4 provides matching lower bounds on the budget that imply the optimality of our procedure. Finally, we present numerical experiments in Section 5 and discuss possible extensions in Section 6.

## 2. Problem formulation and notation

Consider a set of $n$ items, indexed by $[n] = \{1, \ldots, n\}$. Each item is characterized by a feature vector of dimension $d$, where the number of features $d$ may be large. Let $M \in \mathbb{R}^{n \times d}$ be the $n \times d$ matrix such that the $i$-th row of $M$ contains the feature vector of item $i$. We denote the feature vector of item $i$ as $M_{i,\cdot} = (M_{i,1}, \ldots, M_{i,d})$. We assume that the $n$ items are partitioned into two unknown groups, such that items within the same group share the same feature vector. The groups are assumed to be nonempty and non-overlapping. The objective is to recover these two groups.

**Assumption 2.1** (Hidden partition). There exist two distinct vectors $\mu_0 \in \mathbb{R}^d$ and $\mu_1 \in \mathbb{R}^d$, and a non-constant label vector $g \in \{0, 1\}^n$ such that for any item $i \in [n]$,

$$M_{i,\cdot} = \begin{cases} \mu_0 & \text{if } g(i) = 0 \ , \\ \mu_1 & \text{if } g(i) = 1 \ . \end{cases}$$

As is standard in clustering, $(g, \mu_0, \mu_1)$ encodes the same matrix $M$ as $(1 - g, \mu_1, \mu_0)$. Therefore, we assume without loss of generality that $g(1) = 0$ in order to make the label vector $g$ identifiable.

We consider a bandit setting, in which the learner sequentially and adaptively observes noisy entries of the matrix $M$. At each time step $t$, based on passed observations, the learner selects one item $I_t \in [n]$ and one feature $J_t \in [d]$. She then receives $X_t$, a noisy evaluation of $M_{I_t, J_t}$. Conditionally on the pair $(I_t, J_t)$, $X_t$ is an independent sample

drawn from an unknown distribution $\nu_{I_t,J_t}$ with expectation $M_{I_t,J_t}$. The collection of distributions $(\nu_{i,j})_{i,j}$ is referred to as the environment.

We assume that the noise in observations is 1-subGaussian.

**Assumption 2.2** (1-subGaussian noise)**.** For any pair $(i,j) \in [n] \times [d]$, if $X \sim \nu_{i,j}$, then $X - M_{i,j}$ is 1-subGaussian, namely

$$\mathbb{E}\left[\exp(t(X - M_{i,j})\right] \leq \exp\left(t^2/2\right) \quad \forall t \in \mathbb{R} .$$

This subGaussian assumption is standard in the bandit literature (Lattimore & Szepesvári, 2020). It covers, for example, the emblematic case where observations follow Gaussian distributions with variance at most 1, as well as bounded random variables such as those following a Bernoulli distribution on $[0, 1]$. By rescaling, the results can be extended to the case of $\sigma$-subGaussian noise.

We tackle this pure exploration problem in the *fixed confidence setting*, a common framework in the bandit literature (Lattimore & Szepesvári, 2020). In this setting, the learner must decide not only which observations to make but also when to stop. Given a prescribed error probability $\delta$, the learner aims to recover the correct partition of the items with probability at least $1 - \delta$, while minimizing the total number of observations. The learner sequentially collects observations until a stopping time $T$, after which it outputs an estimated label vector $\hat{g} \in \{0, 1\}^n$ (satisfying $\hat{g}(1) = 0$). The total number of observations, given by $T$, is referred to as the *budget* of the procedure. Formally, $T$ is a stopping time with respect to the natural filtration associated to the sequential model.

For any confidence level $\delta$, we say then that a procedure $\mathcal{A}$ is $\delta$-PAC (Probably Approximately Correct) if

$$\mathbb{P}_{\mathcal{A},\nu}(\hat{g} = g) \geqslant 1 - \delta ,$$

where $\mathbb{P}_{\mathcal{A},\nu}$ denotes the probability distribution induced by the interaction between the environment $\nu$ and the algorithm $\mathcal{A}$.

The performance of a $\delta$-PAC algorithm is evaluated through its budget $T$, which should be as small as possible. In this paper, we derive upper bounds on $T$ that hold with high probability—typically on an event of probability at least $1 - \delta$, under which the algorithm returns a correct clustering.

We introduce two key quantities for analyzing the problem. The *gap vector* is defined as

$$\Delta := \mu_1 - \mu_0 \in \mathbb{R}^d ,$$

which naturally captures the difficulty of the clustering task. By assumption, we have $\Delta \neq 0$ so that there is exactly two disjoint groups. The smaller the norm of $\Delta$ is, the more

challenging the estimation of $g$ becomes. In particular, we analyze the complexity of the clustering task with respect to the entry of the gap vector $\Delta$ ordered by decreasing absolute value, namely $\left|\Delta_{(1)}\right| \geq \left|\Delta_{(2)}\right| \geq \cdots \geq \left|\Delta_{(d)}\right|$.

Most intuitions behind our method rely on the sparse setting, where the two groups differ in exactly $s$ entries with a constant gap $h$. This corresponds to the case where the gap vector $\Delta$ has exactly $s$ nonzero entries equal to $h > 0$ and $d - s$ entries equal to 0. The smaller the sparsity level $s$, the more entries must be explored to detect a discriminative feature. The smaller the magnitude $h$, the more budget is required to distinguish between the two groups.

Besides, we define $\theta$ the balancedness of the partition $g$, that is the proportion of arms in the smallest group

$$\theta := \frac{1}{n}\sum_{i=1}^{n}\mathbb{1}(g(i) = 0) \wedge \frac{1}{n}\sum_{i=1}^{n}\mathbb{1}(g(i) = 1) .$$

Intuitively, the smaller $\theta$ is, the more unbalanced the partition is, and the more difficult it is to discover two items of distinct groups. For identifiability reasons, we assumed in Assumption 2.1 that the groups are nonempty – which implies in particular that $n \geqslant 2$, but also that $\theta \in [1/n; 1/2]$.

## 3. Algorithms

### 3.1. Introduction to our method

We introduce briefly our method, which contains two steps.

First, we fix arbitrary the item $r_0 = 1$ as a representative of the first group[2]. Then, we aim to identify a second representative item $r_1 \in [n]$ that belongs to the other group, and a feature $j \in [d]$ such that $M_{r_1,j}$ differs from $M_{r_0,j}$ significantly, that is such that $|M_{r_1,j} - M_{r_0,j}|$ is large. Our method balances the budget spent on identifying such discriminative feature, with the budget required for classifying all items based on this feature. Improving the budget compared to non-active settings requires over-sampling certain rows , which we refer to as representatives, following (Thuot et al., 2025). This step is crucial for accurately estimating some entries of the vectors $\mu_0$ and $\mu_1$ and, ultimately, for accelerating the clustering task.

Importantly, if we detect an entry $(r_1, j)$ in the matrix where the gap $|M_{r_1,j} - M_{r_0,j}|$ is sufficiently large, we obtain two complementary pieces of information. This naturally leads us to organize the clustering task as a two-step procedure:

1. If we can test that $|M_{r_1,j} - M_{r_0,j}| > 0$, then item $r_1$ can serve as a representative of the second group. Algorithm 2 identifies such an item $r_1$ with high probability. Once the two representatives $r_0$ and $r_1$ are known, we

---

[2]By symmetry, any randomly selected row could serve the same purpose.

allocate a significant portion of the budget to these items in order to identify a discriminative feature in the gap-vector $\Delta$.

2. If we identify a feature $j$ such that $|M_{r_1,j} - M_{r_0,j}|$ not only differs from zero but also exceeds a certain threshold—specified later—then feature $j$ is deemed sufficiently discriminative, and we concentrate the classification budget on this feature. We then estimate $|\Delta_j| = |M_{r_0,j} - M_{r_1,j}|$ with samples from entries $(r_0, j)$ and $(r_1, j)$, and classify the remaining items with a budget of order $O\left(\frac{n}{\Delta_j^2} \log(n/\delta)\right)$, uniformly allocated over items in the $j$-th column of $M$. This second step is detailed in Algorithm 3.

### 3.2. Warm-up: adaptation of Sequential Halving

As a subroutine, we introduce CSH for CompareSequentialHalving, which is detailed in Algorithm 1. It is a variant of the Sequential Halving (SH) algorithm, introduced in (Karnin et al., 2013), which is very similar to Bracketing Sequential Halving described in (Zhao et al., 2023, Alg. 3). Building on recent advances in the analysis of SH applied to various best arm identification problems (Zhao et al., 2023), we analyze the performance of the method in a specific problem that we introduce – we provide an explicit guarantee for CSH in Lemma D.1.

The algorithm takes 4 entries $r_0, I, L, T$. Given a fixed item $r_0 \in [n]$ and a subset of items $I \subset [n] \setminus r_0$, CSH outputs an item $i \in I$ and a feature $j \in [d]$ for which the absolute difference $|M_{i,j} - M_{r_0,j}|$ is as large as possible. Each time we run CSH, we allocate a budget $T^3$, that the algorithm can fully spend. That is, CSH operates under a fixed budget constraint.

Following the literature on best arm identification with multiple good arms (Berry et al., 1997; Katz-Samuels & Jamieson, 2020; Jamieson et al., 2016; De Heide et al., 2021), we incorporate a sub-sampling mechanism. Initially, CSH selects randomly $S_0$, a subset of $2^L$ entries from $I \times d$, where $L$ is a parameter specifying the sub-sampling size. Sequential Halving is then applied exclusively to the selected subset, rather than to the entire matrix. The optimal choice for $L$ balances two factors. If $L$ is small, the algorithm concentrates more budget per entry, enabling the detection of smaller gaps. If $L$ is large, we increase the likelihood of including in $S_0$ a significant proportion of good entries, ensuring that the quality of the remaining entry is not limited by an unlucky draw.

We employ CSH as a subroutine in both Algorithm 2 and

---

³Actually, row $r_0$ is sampled half of the time, so that each sample from $\nu_{i,j}$ is compared to a new one from $\nu_{r_0,j}$. Any more refined book-keeping of samples from row $r_0$ would at best improve the budget by 2.

---

Algorithm 3. In Algorithm 2, we explore the entire matrix in order to detect a row $r_1$ that differs from the first row $r_0 = 1$, for this we use $I = [n] \setminus r_0$. In Algorithm 3, we focus the exploration on two rows $r_0 = 1$ and $r_1$ ($I = \{r_1\}$), aiming at detecting a feature that best separates the two groups represented by $r_0$ and $r_1$.

---

**Algorithm 1** CompareSequentialHalving (CSH)

**Require:** $r_0$ an item, $I \subset [n] \setminus r_0$ subset of items, $L$ number of halving steps, $T \geq 2^{L+2}$ budget
**Ensure:** a couple $(i, j) \in I \times [d]$
1: selects $S_0 \leftarrow \{(i_1, j_1), (i_2, j_2), \ldots, (i_{2^L}, j_{2^L})\}$ uniformly with replacement from $I \times [d]$
2: **for** $l = 1, \ldots, L$ **do**
3:     $\tau_l \leftarrow \left\lfloor \frac{T}{2^{L-l+2}L} \right\rfloor$
4:     **for** $(i, j) \in S_{l-1}$ **do**
5:         draw $\begin{cases} X_{r_0,j}^{(1)}, \ldots, X_{r_0,j}^{(\tau_l)} \sim^{\text{i.i.d.}} \nu_{r_0,j} \\ X_{i,j}^{(1)}, \ldots, X_{i,j}^{(\tau_l)} \sim^{\text{i.i.d.}} \nu_{i,j} \end{cases}$
6:         store $\widehat{D}_{i,j} \leftarrow \frac{1}{\tau_l} \sum_{u=1}^{\tau_l} \left(X_{i,j}^{(u)} - X_{r_0,j}^{(u)}\right)$
7:     **end for**
8:     keep in $S_l$ indices $(i, j) \in S_{l-1}$ corresponding to the $2^{L-l}$ largest $\left|\widehat{D}_{i,j}\right|$
9: **end for**
10: return $(i, j) \in S_L$

---

### 3.3. First step: CandidateRow

We start our procedure by solving a sub-problem which consists on detecting an item $r_1$ which is not in the same group as the prefixed representative $r_0 = 1$. We perform this step in the following Algorithm 2. The guarantees of Algorithm 2 are proved in Appendix E and are gathered in Proposition E.1.

In Algorithm 2, we perform multiple runs of the CHS subroutine, iteratively increasing the budget $T_k = 2^{k+1}$ allocated for each run. For a given run of CSH$(r_0, [n], L, T_k)$, we obtain an entry $(i, j)$ (Line 4). In Line 5, we use the same amount of observations $2^{k+1}$ to estimate the gap $|M_{i,j} - M_{r_0,j}|$. Finally, in Line 8, we perform a test based on a Hoeffding's bound to decide whether $|M_{i,j} - M_{r_0,j}| > 0$ or not. If at some point, this test concludes, Algorithm 2 outputs $r_1 = i$. The threshold chosen in the stopping condition from Line 6 is designed to assure that with a probability larger than $\delta$, then the selected item $r_1$ belongs to another group as $r_0 = 1$.

In Line 3, we chose $L_{\max}$ as

$$L_{\max} := \left\lceil \log_2 \left(16dn \log \left(\frac{4 \log(8nd)}{\delta}\right)\right)\right\rceil ,$$

which corresponds to the sub-sampling budget required, according to Lemma D.1, when $\theta = 1/n$ takes the smallest

---

**Algorithm 2** `CandidateRow` (CR)

**Require:** confidence parameter $\delta > 0$, item $r_0$
**Ensure:** row index $r_1 \in [n]$
1: initialize $r_1 \leftarrow 0$, $k \leftarrow 1$
2: **while** $r_1 = 0$ **do**
3:    **for** $1 \leq L \leq L_{\max}$ such that $L \cdot 2^L \leq 2^{k+1}$ **do**
4:       $(i,j) \leftarrow \mathrm{CSH}([n], L, 2^{k+1})$
5:       draw $\begin{cases} X_{r_0,j}^{(1)}, \ldots, X_{r_0,j}^{(2^k)} \overset{\text{i.i.d.}}{\sim} \nu_{r_0,j} \\ X_{i,j}^{(1)}, \ldots, X_{i,j}^{(2^k)} \overset{\text{i.i.d.}}{\sim} \nu_{i,j} \end{cases}$
6:       **if** $\left| \sum_{t=1}^{2^k} \left( X_{i,j}^{(t)} - X_{r_0,j}^{(t)} \right) \right| >$
        $\sqrt{4 \cdot 2^k \log \left( \left( \frac{k^3}{0.15\delta} \right) \right)}$ **then**
7:          $r_1 \leftarrow i$
8:       **end if**
9:    **end for**
10:   $k \leftarrow k + 1$
11: **end while**

---

possible value.

From Lemma D.1, we know that for $\Delta_{(s)}^2 \leq 128L^2$, for some constant $c$, if the condition

$$T_k = 2^{k+1} \geqslant cL_{\max}^3 \frac{d \left( \log(1/\delta) + \log\log(nd) \right)}{\theta s \Delta_{(s)}^2} \ ,$$

holds, then, with high probability, there exists $L \leqslant L_{\max}$ such that $\mathrm{CHS}(r_0, [n], L, T_k)$ outputs a pair $(i,j)$ with $|M_{i,j} - M_{r_0,j}| \geq |\Delta_{(s)}|/2$. Besides, under this budget condition, the termination condition from Line 8 will be reached w.h.p. This condition is especially true for the sparsity $s \in [d]$, where the inequality above is tightest.

As in (Jamieson et al., 2016; Saad et al., 2023), the exponential grid $T_k = 2^k$ allows us to adapt the strategy, and reach a budget that scales up to log terms as $O\left( \min_{s \in [d]} \frac{d}{\theta s \Delta_{(s)}^2} \log(1/\delta) \right)$, even without prior knowledge of this quantity by the learner.

Interestingly, we can relate this quantity to the $l^2$ norm of $\Delta$. For that, we define

$$s^* \in \mathrm{argmax}_{s \in [d]} s \cdot \Delta_{(s)}^2 \ . \tag{1}$$

This quantity, $s^*$, appears as an effective sparsity parameter, as observed in signal detection contexts. Actually, the following bound holds

$$\max_{s \in [d]} s \cdot \Delta_{(s)}^2 \leq \|\Delta\|_2^2 \leq \log(2d) \max_{s \in [d]} s \cdot \Delta_{(s)}^2 \ . \tag{2}$$

Finally, we prove that Algorithm 2 outputs an item $r_1$ which belongs to the second group with a probability larger than $1 - \delta$, using a budget that is smaller, up to logarithmic terms, than the quantity $\frac{d}{\theta \|\Delta\|_2^2} \log(1/\delta)$. We will see in Theorem 4.1 that this bound is optimal for both $\theta$ and $\Delta$.

## 3.4. Second step: `ClusterByCandidates`

Consider the high probability event on which, after the first step of our procedure, Algorithm 2 provides an item $r_1 \in [n]$ such that $M_{r_1,\cdot} \neq M_{r_0,\cdot}$. Our next goal is to select a feature $j$ such that $|\Delta_j|$ is large enough to allow a quick classification of each of the $n$ items. We propose the procedure Algorithm 3 which shares a similar structure with Algorithm 2. The guarantees of Algorithm 3 are proved in Appendix E, in Proposition F.1.

We begin by performing several runs of `CSH`, with $I = \{r_1\}$ in order to detect large entries in the (absolute) gap vector $\Delta$. As in Algorithm 2, we perform `CSH` with an increasing sequence of budget $T_k = 2^{k+1}$, and for each budget $T_k$, we test all possible sub-sampling sizes $L = 1, \ldots, \tilde{L}_{\max}$. In Line 3, we define

$$\tilde{L}_{\max} := \left\lceil \log_2 \left( 16d \log \left( \frac{4\log(8d)}{\delta} \right) \right) \right\rceil \ ,$$

as the maximum sub-sampling size needed to detect a signal when the sparsity level $s = 1$ is the smallest.

In Line 4, we call $\mathrm{CSH}(r_0, \{r_1\}, L, 2^{k+1})$. We obtain a feature $j$ and then estimate $|\Delta_j| = |M_{r_0,j} - M_{r_1,j}|$ in Line 6. For that, we take $\lfloor 2^k/n \rfloor$ samples from $\nu_{r_0,j}$ and $\nu_{r_1,j}$, and compute the sum of differences $\hat{D} = \sum_{t=1}^{\lfloor 2^k/n \rfloor} \left( X_{r_1,j} - X_{r_0,j} \right)$. We deduce a high probability lower bound $|\hat{\Delta}|_j = \frac{1}{\lfloor 2^k/n \rfloor}(\hat{D} - \epsilon) \leqslant |\Delta_j|$, where $\epsilon$ is defined in Line 7. Based on $|\hat{\Delta}|_j$, we can classify (with high probability) each item by sampling the $j$-th feature $O\left( \frac{1}{\hat{\Delta}^2} \log(n/\delta) \right)$. We can then assess whether the classification budget required for feature $j$ is feasible given the budget $T_k$. If $T_k \leqslant \frac{cn}{|\hat{\Delta}|_j^2} \log(n/\delta)$, it seems that with feature $j$, the classification budget exceeds $T_k$, we discard this feature and repeat `CSH` with larger sub-sampling size $L$ or budget $T$.

We now bound the budget of our procedure thanks to Lemma D.1. Assume that $\Delta_{(s)}^2 \leq 128L^2$. If it holds for some constant $c$ that

$$T_k = 2^{k+1} \geqslant cL_{\max}^3 \frac{d \left( \log(1/\delta) + \log\log(d) \right)}{s \Delta_{(s)}^2} \ ,$$

then, there exists $L \leqslant \tilde{L}_{\max}$ such that $\mathrm{CHS}(r_0, r_1, L, T_k)$ outputs a feature $j$ such that $|M_{r_0 j} - M_{r_1 j}| \geq |\Delta_{(s)}|/2$. If we also have $T_k \geqslant c\frac{n}{\Delta_{(s)^2}} \log(n/\delta)$, then the algorithm stops, otherwise it would continue sampling.

Overall, we prove that the total budget of the procedure, up to logarithmic factors, is no more than

$$\min_{s \in [d]} \left( \frac{d}{s} + n \right) \left( \frac{1}{\Delta_{(s)}^2} + 1 \right) \log(1/\delta) \ .$$

**Algorithm 3** `ClusterByCandidates (CBC)`

---
**Require:** confidence $\delta > 0$, representative items $r_0, r_1 \in [n]$
**Ensure:** labels $\hat{g} \in \{0,1\}^n$
1: $\hat{g} \leftarrow (0, \ldots, 0)^T \in \{0,1\}^n$, $k \leftarrow \lceil \log_2(n) \rceil$
2: **while** True **do**
3:     **for** $1 \leq L \leq \tilde{L}_{\max}$ such that $L \cdot 2^L \leq 2^{k+1}$ **do**
4:         $j \leftarrow \texttt{CSH}(r_0, r_1, L, 2^{k+1})$
5:         draw $\begin{cases} X_{r_0,j}^{(1)}, \ldots, X_{r_0,j}^{(\lfloor 2^k/n \rfloor)} \sim^{\text{i.i.d.}} \nu_{r_0,j} \\ X_{r_1,j}^{(1)}, \ldots, X_{r_1,j}^{(\lfloor 2^k/n \rfloor)} \sim^{\text{i.i.d.}} \nu_{r_1,j} \end{cases}$
6:         $\hat{D} \leftarrow \sum_{t=1}^{\lfloor 2^k/n \rfloor} (X_{r_1,j} - X_{r_0,j})$
7:         $\varepsilon \leftarrow \sqrt{4 \cdot \lfloor 2^k/n \rfloor \log(nk^3/0.15\delta)}$
8:         **if** $\left| \hat{D} \right| \geq 3 \cdot \varepsilon$ **then**
9:             **for** $i \in [n]$ **do**
10:                draw $\begin{cases} X_{r_0,j}^{(1)}, \ldots, X_{r_0,j}^{(\lfloor 2^k/n \rfloor)} \sim^{\text{i.i.d.}} \nu_{r_0,j} \\ X_{i,j}^{(1)}, \ldots, X_{i,j}^{(\lfloor 2^k/n \rfloor)} \sim^{\text{i.i.d.}} \nu_{i,j} \end{cases}$
11:                $\hat{D}_i \leftarrow \sum_{t=1}^{\lfloor 2^k/n \rfloor} (X_{i,j} - X_{r_0,j})$
12:                $\hat{g}(i) \leftarrow \mathbb{1}\left( \left| \hat{D}_i \right| \geq \varepsilon \right)$
13:             **end for**
14:             **output** $(\hat{g}(1), \ldots, \hat{g}(d))$
15:         **end if**
16:     **end for**
17: **end while**

---

### 3.5. Main Algorithm

To obtain a complete clustering procedure that is adaptive to $\Delta$ and $\theta$, one simply has to combine Algorithm 2 and Algorithm 3. The overall clustering procedure is then given in Algorithm 4.

**Algorithm 4** `BanditClustering`

---
**Require:** confidence parameter $\delta > 0$
**Ensure:** labels $\hat{g} \in \{0,1\}^n$
1: Fix $r_0 = 1$
2: $r_1 \leftarrow \texttt{CR}(\delta/2, r_0)$
3: $\hat{g} \leftarrow \texttt{CBC}(\delta/2, r_0, r_1)$

---

**Theorem 3.1.** *For $\delta \in (0, 1/2e)$, consider Algorithm 4 with entry $\delta$. Define*

$$H := \frac{d}{\theta}\left( \frac{1}{\|\Delta\|^2} + \frac{1}{s^*} \right) + \min_{s \in [d]} \left( \frac{d}{s} + n \right) \left( \frac{1}{\Delta_{(s)}^2} + 1 \right),$$
(3)

*where $s^*$ is the effective sparsity defined in (2).*

*With a probability of at least $1 - \delta$, Algorithm 4 returns*

$\hat{g} = g$ *with a budget of at most*

$$T \leqslant \tilde{C} \cdot \log\left( \frac{1}{\delta} \right) \cdot H,$$

*where there exists a numerical constant $C$, and an index $\tilde{s} = s^* \vee (\lceil d/n \rceil \wedge |\{j \in [d], \Delta_j \neq 0\}|)$, such that $\tilde{C}$ is a logarithmic factor smaller than*

$$C \cdot (\log\log(1/\delta) \vee 1)^4$$
$$\cdot \log(dn)^5 \log(d)(\log_+ \log(1/\Delta_{(\tilde{s})}^2) \vee 1).$$

In order to understand the motivation behind our complexity $H$, we write our main theorem in the sparse setting – $\Delta$ is $s$-sparse with constant magnitude $h$.

**Corollary 3.2.** *For $\delta \in (0, 1/e)$ and $\Delta \in \{0, h\}^d$ with $0 < h < 1$, with a probability of at least $1 - \delta$, Algorithm 4 returns $\hat{g} = g$ with a budget of at most*

$$\tilde{C} \cdot \log(1/\delta) \cdot \left( \frac{d}{\theta \|\Delta\|^2} + \frac{n}{h^2} \right),$$

*where $\tilde{C}$ is a logarithmic factor smaller than*

$$C \cdot (\log\log(1/\delta) \vee 1)^4 \cdot \log(dn)^5 (\log_+ \log(1/h^2) \vee 1),$$

*with a numerical constant $C > 0$.*

Compared to the lower bound in $Theorem$ 4.1, we prove that our procedure is optimal when the gap vector is $s$-sparse with a constant magnitude $h$. For a general gap vector $\Delta$, we have good reasons to think that understanding the optimality of this trade-off for this simple example allows us to understand (at least intuitively) the optimality for general vectors.

We interpret $H$ in (3) as a non asymptotic sampling complexity, which depends on the instance-specific parameters of our model $\theta$, $\Delta$, $n$, and $d$. The complexity $H$ can be decomposed as two terms:

**First Term:** $\frac{d}{\theta \|\Delta\|^2} \log(1/\delta)$, which correspond to the budget used to identify an item belonging to the second group. In the sparse setting, it scales as $\frac{d}{\theta s} \times \frac{1}{h^2}$, which is necessary, as we need to explore at least $\frac{d}{\theta s}$ entries to find a non-zero entry, and we need at least $1/h^2$ samples from each of these entries to decide if it is equal to 0 or not with a constant probility of error.

**Second Term:** $\min_{s \in [d]} \left( \frac{d}{s} + n \right) \left( \frac{1}{\Delta_{(s)}^2} + 1 \right)$. This term represents the best trade-off between the exploration of the gap vector $\Delta$ and its exploitation for clustering. Indeed, the term $\frac{n}{\Delta_{(s)}^2} \log(1/\delta)$ is the price for clustering if we use a feature with a gap $|\Delta_{(s)}|$ whereas $\frac{d}{s\Delta_{(s)}^2} \log(1/\delta)$ corresponds to the price for identifying a feature with a gap at least $|\Delta_{(s)}|$.

Define the effective sparsity $\tilde{s}$ for which the minimum holds, and define the effective magnitude as $\Delta_{(\tilde{s})}$. In the sparse setting, this is exactly the sparsity level. Intuitively, we can argue that entries significantly larger than $\Delta_{(\tilde{s})}$ are too rare to be detected (otherwise, $\tilde{s}$ would be smaller), and entries much smaller than $\Delta_{(\tilde{s})}$ are too weak to be used for classification with a budget $H$. Our insight is that the problem is as hard as if the gap vector were $\tilde{s}$-sparse with a constant magnitude $\Delta_{(s^*)}$, a setting where we have matching lower and upper bounds (see Corollary 3.2 and Theorem 4.1), leading to our complexity.

Finally, we mention that the remaining term $d/(\theta s^*)$ in $H$ only dominates in the very specific setting where the non-zero entries of $\Delta$ are really large so that $\|\Delta^2\| \geq s^*$.

## 4. Lower bounds

In this section, we provide a lower bound on the budget of any $\delta$-PAC algorithm. The bound is instance-dependent, meaning it holds for a specific problem instance defined by the matrix $M$. We establish this result by constructing a family of alternative environments, each obtained by slightly modifying the original matrix $M$. We then prove that any algorithm with a budget that is too small cannot perform well simultaneously across all these environments. For the lower bound, we consider Gaussian environments, for which Assumption 2.2 holds.

We define $\mathcal{E}_{per}(M)$ as the set of Gaussian environments constructed from $M$ by permuting its rows and columns. Without loss of generality, we assume that $\mu_0 = 0$ and $\mu_1 = \Delta$. Formally, an environment $\tilde{\nu} \in \mathcal{E}\mathrm{per}(M)$ is defined using a permutation $\sigma$ of $[n]$ and a permutation $\tau$ of $[d]$ as follows:

$$\tilde{\nu}_{i,j} = \begin{cases} \mathcal{N}(0,1) & \text{if } g(\sigma(i)) = 0 \\ \mathcal{N}(\Delta_{\tau_j}, 1) & \text{if } g(\sigma(i)) = 1 \end{cases}, \qquad (4)$$

where $g \in \{0,1\}^n$ denotes the unknown labels associated to matrix $M$. Intuitively, permuting the rows and columns of $M$ accounts for the fact that (a) the target labels $g$ are not available to the learner, and (b) the structure of the gap vector $\Delta$ is also unknown.

**Theorem 4.1.** *Fix $\delta \in (0, 1/4)$. Assume that $\mathcal{A}$ is $\delta$-PAC for the clustering task, then, there exists $\tilde{\nu} \in \mathcal{E}_{per}(M)$ such that the $(1 - \delta)$-quantile of the budget of algorithm $\mathcal{A}$ is bounded as follows*

$$\mathbb{P}_{\mathcal{A},\tilde{\nu}} \left( T \geq \frac{2(n-2)}{\Delta_{(1)}^2} \log \left( \frac{1}{4.8\delta} \right) \vee \frac{2d}{\theta \|\Delta\|_2^2} \log \frac{1}{6\delta} \right) \geq \delta$$
$$(5)$$

The lower bound contains two terms. The first term scales as $\frac{d}{\theta \|\Delta\|^2} \log(1/\delta)$, and can be interpreted as the budget required to identify one item from each group, while adapting to the unknown structure of the gap vector $\Delta$. This term matches, up to logarithmic factors, the budget incurred in the first step of our algorithm for identifying a relevant item. In particular, it implies that the first step from Algorithm 2 is optimal.

The second term scales as $\frac{n}{\Delta_{(1)}^2} \log(1/\delta)$, and take into account the difficulty of clustering all items once a discriminative feature is identified. Specifically, if the most informative feature is provided by an oracle—i.e., a feature index $j \in [d]$ such that $|\Delta_j| = \Delta_{(1)}$ is maximal—then the problem reduces to performing $n$ independent Gaussian hypothesis tests of the form $H_0 : X \sim \mathcal{N}(0,1)$ versus $H_1 : X \sim \mathcal{N}(\Delta_{(1)}, 1)$.

In the case where the gap vector $\Delta$ takes two values, this lower bound matches, up to poly-logarithmic terms, the upper bound from Corollary 3.2. In summary, when $\Delta$ only takes two values, our budget is optimal with respect to $d$, $n$, $\theta$ and $\log(1/\delta)$. For more general $\Delta$, we conjecture that the trade-off in $H$ in (3) is optimal and unavoidable.

## 5. Experiments

We support our theoretical results with numerical experiments on synthetic data. In the first experiment, we investigate the sample complexity of our algorithm in sparse regimes. We compare its performance to a uniform sampling baseline, where each of the $n \times d$ entries of the matrix $M$ is sampled an equal number of times, and the resulting data is clustered using the $K$-means algorithm. In the second experiment, we evaluate the `BanditClustering` algorithm under fixed sparsity while simultaneously increasing the parameters $n$ and $d$, and we compare different choices of the confidence parameter $\delta > 0$. In a third experiment, we compare `BanditClustering` to `Adaptive Clustering` (Ariu et al., 2024) and see how a growing number of features impacts both algorithms performances. We defer a fourth experiment to Appendix B, which illustrates the influence of $\theta$ on the budget of `BanditClustering`. As shown in Corollary 3.2, the effect of $1/\theta$ is comparable to that of the sparsity $s$ discussed in the first experiment.

The code we used for the simulations is available in a GitHub repository[4].

**Experiment 1** In this experiment, we consider a small number of items ($n = 20$) and a large number of features ($d = 1000$). For $s \in 1, \ldots, d$, define the vector $\Delta^s = (\underbrace{h_s, \ldots, h_s}_{s \text{ times}}, 0, \ldots, 0)$, where $h_s = 15/\sqrt{s}$. This choice ensures that $\Delta^s$ is $s$-sparse and satisfies $\|\Delta^s\|_2 = 15$ for any

---

[4]https://github.com/grafmaxi/bandit_two_clusters

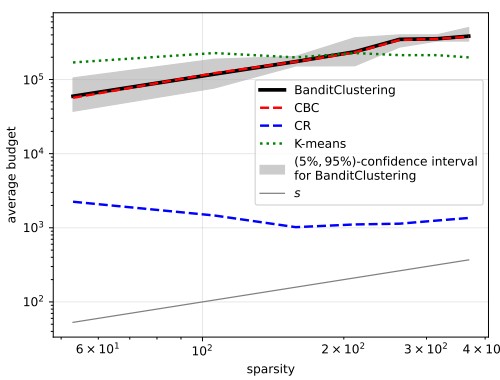

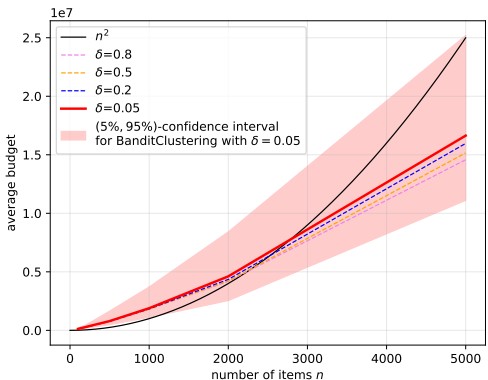

*Figure 1.* Different budgets for Experiment 1, depending on the sparsity of $\Delta^s$.

*Figure 2.* Different budgets for Experiment 2, depending on the dimensionality of the problem $n$ and $d = 10 \cdot n$.

$s$. We construct the matrix $M^s$ with half of its rows equal to 0 and the other half equal to $\Delta^s$, and sample observations as $\nu_{i,j} \sim \mathcal{N}(M^s_{i,j}, 1)$. We examine the behavior of our algorithms as $s$ varies, which causes the magnitude $h_s$ to vary accordingly.

We report separately the budgets required by the two steps of our algorithm. For $\delta = 0.8$, we run each of the algorithms CR($\delta/2$), CBC($\delta/2, i^*$) and BanditCluster($\delta$) a total of $\kappa = 5000$ times. We provide to CBC a representative item $i^*$ in the second group as an oracle. In this setting, the observed error rate of BanditClustering remains close to 0.01 for all values of $s$. We depict the average budgets required by BanditClustering($\delta$), considering only the runs where the first step CR returns a valid candidate row (otherwise, we emergency stop the algorithm). We also show the average budget required by CR($\delta/2$) and CBC($\delta/2, i^*$) in Figure 1. The figure includes the $(0.05, 0.95)$ quantiles across simulations.

As a benchmark, we compare our algorithm to a strategy that samples uniformly all entries of $M^s$, and then applies the KMeans algorithm from the Scikit-learn library (Pedregosa et al., 2011). Given a budget $T$, we sample $\tau = \lfloor T/nd \rfloor$ observations $X^{(t)}_{i,j} \sim^{\text{i.i.d.}} \mathcal{N}(M^s_{i,j}, 1)$ per entry, and compute $\bar{X}_{i,j} = \frac{1}{\tau} \sum_{t=1}^{\tau} X^{(t)}_{i,j}$. We then cluster the items by performing the $K$-means algorithm with the vectors $(\bar{X}_{1j})_{j=1,\dots,d}, \dots, (\bar{X}_{nj})_{j=1,\dots,d}$. For each sparsity $s$, the budget $T$ is turned by an oracle, with a grid search, so that the observed error rate after $\kappa = 5000$ runs is below 0.01. The resulting budgets are reported in Figure 1.

From Figure 1 we observe that in the sparse regime, our algorithm requires fewer observations than the uniform sampling approach used as baseline. Moreover, the budget required for CR appears to be mostly independent of the sparsity level $s$. In this setting, the overall sample com-

plexity of BanditClustering is mainly driven by the cost of the CBC step, which grows approximately linearly with $s$. These empirical dependencies on $s$ align with our theoretical results, where the budget of CR in this case is, up to polylogarithmic factors, of order $d/\theta \|\Delta\|_2^2$, which is constant in $s$ for fixed $\|\Delta\|_2$, while CBC requires a budget of order $n/(\Delta_i^s)^2 \sim n \cdot s$.

**Experiment 2** In this experiment, we consider matrices of increasing size, with $n \in 100, 200, 500, 1000, 2000, 5000$ and $d = 10 \cdot n$, so that the number of features grows proportionally with the number of items. For each value of $n$, we define the vector $\tilde{\Delta}^{(n)} \in \mathbb{R}^d$ as $\tilde{\Delta}^{(n)} = (\underbrace{5, \dots, 5}_{10 \text{ times}}, \underbrace{0, \dots, 0}_{d-10 \text{ times}}) \in \mathbb{R}^d$. We construct a matrix $M^{(n)}$ with $n/2$ rows equal to 0 and $n/2$ rows equal to $\tilde{\Delta}^{(n)}$, and we add Gaussian noise with unit variance. For each $\delta \in 0.8, 0.5, 0.2, 0.05$, we run BanditClustering($\delta$) over $\kappa = 5000$ independent trials. In Figure 2, we report the average budget required by BanditClustering for each configuration. For $\delta = 0.05$, we additionally report the 5th and 95th percentiles across the simulations.

If we allocate a budget smaller than $5n^2 = nd/2$ uniformly at random across the $nd$ entries of $M$, then on average, each item $i$ will have $d/2$ unobserved features. As both $n$ and $d$ increase, the probability that some item $i$ is only sampled on coordinates $j$ such that $\tilde{\Delta}_j^{(n)} = 0$ tends to one, rendering accurate clustering impossible. By contrast, Figure 2 shows that the budget required by our algorithm scales linearly with $n$. This matches the bounds from Corollary 3.2, which implies that when $d = 10 \cdot n$ and the parameters $\theta$, $s$, and $h$ are fixed, the total budget required (up to polylogarithmic factors) is of order $n$.

**Experiment 3** In this third experiment, we compare our algorithm BanditClustering with the Adaptive

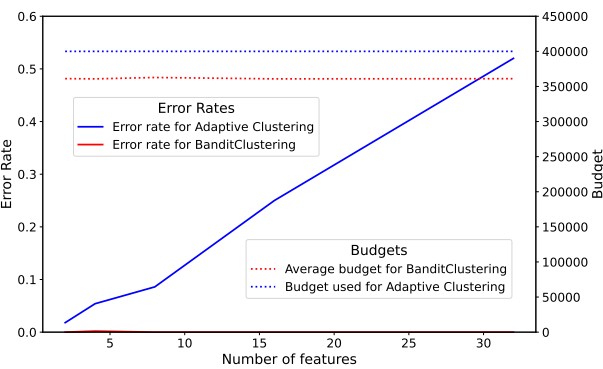

*Figure 3.* Comparison of the performance of `BanditClustering` and `Adaptive Clustering` in Experiment 3, depending on the number of features $d_\gamma$.

`Clustering` algorithm introduced as Algorithm 2 in (Ariu et al., 2024). We fix the number of items to $n = 30$ and vary the number of features as $d_\gamma = 2^\gamma$ with $\gamma = 1, \ldots, 5$. We set $\theta = 0.5$, and for each $d_\gamma$, we define $s_\gamma = d_\gamma/2$ features $j$ such that $M_{i,j} = 0.25$ for items in the first cluster and $M_{i,j} = 0.75$ for items in the second cluster, yielding $h = 0.5$. For the remaining features, we set $M_{i,j} = 0.5$, independent of the item's cluster. Observations are sampled from Bernoulli distributions: $\nu_{i,j} = \text{Bern}(M_{i,j})$. We apply `BanditClustering` with $\delta = 0.8$ over $\kappa = 500$ runs. We then ran the `Adaptive Clustering` algorithm using the MATLAB code provided by (Ariu et al., 2024), under the same setting and a fixed budget of $T = 400,000$, also with $\kappa = 500$ runs. In Figure 3, we compare this fixed budget to the average budget used by `BanditClustering`, and report the corresponding error rates as a function of $d_\gamma$.

As the number of features increases, we observe that the error of the fixed-budget algorithm `Adaptive Clustering` also increases. In contrast, the fixed-confidence algorithm `BanditClustering` maintains a consistently low error, and its average required budget remains largely unchanged. This behavior aligns with Corollary 3.2, as both $d_\gamma/\theta\|\Delta\|^2$ and $n/h^2$ are constant in our experimental setup. While our algorithm performs better in this specific scenario, it is important to note that the problem addressed in (Ariu et al., 2024) is more complex than the clustering task we consider. We believe their algorithm could be adapted to our setting; however, it remains unclear whether the influence of the number of features on the error rate can be mitigated.

## 6. Discussion

**Comparison to other active clustering settings and batch clustering.** In this work, we consider a bandit clustering setting where the learner can adaptively sample each item-

feature pair. This contrasts with (Yang et al., 2024; Thuot et al., 2025; Yavas et al., 2025) where the authors have to sample all the features for each item and cannot focus on most relevant features. Rewriting their results in our setting, the optimal budget for the latter problem is, up to poly-logarithmic terms, of the order of

$$\frac{nd\log(1/\delta)}{\|\Delta\|^2} + \frac{d^{3/2}\sqrt{n\log(1/\delta)}}{\|\Delta\|^2} .$$

Comparing this with our main result (Theorem 3.1), we first observe that the ability to adaptively select features allows to remove the so-called high-dimensional terms $d^{3/2}\sqrt{n\log^{1/2}(1/\delta)}/\|\Delta\|^2$ that occurs when the number of features is large - $d \geq n\log(1/\delta)$. Second, the adaptive queries allow to drastically decrease the budget in situation where the vector $\Delta$ contains a few large entries so that a few feature are especially relevant to discriminate. To illustrate this, consider e.g. a setting as in Corollary 3.2 where $\Delta \in \{0, h\}^d$ takes $s$ non-zero values where the partition is balanced so that $\theta = 1/2$. Then, our budget is of the order of

$$\log(1/\delta)\left[\frac{d}{sh^2} + \frac{n}{h^2}\right] ,$$

which represents a potential reduction by a factor $n \wedge \frac{d}{s}$ compared to (Yang et al., 2024; Thuot et al., 2025).

**Extension to a larger number of groups.** Throughout this work, we have assumed that the items are clustered into $K = 2$ groups. However, our algorithms can be used as subroutines to address the case where $K > 2$. In Appendix C, we provide an algorithm that handles this extension, along with a (non-optimal) budget that scales with $K^2$. A key challenge in achieving optimality in this setting is determining whether the algorithm should focus on all $K(K-1)/2$ pairwise discriminative features or on a smaller, more informative subset. It is significantly more difficult to devise a strategy that adapts optimally to the relative positions of the centers of the $K$ groups. We leave this question for future work.

**Extension to heterogeneous groups.** We also assumed throughout this work that all items within a group are perfectly similar, meaning their corresponding mean vectors $\mu_i$ are equal. This assumption could be relaxed by allowing the $\mu_i$'s within a group to be close, but not necessarily identical. For instance, suppose we have prior knowledge that, for any feature $j \in [d]$, the within-group variation satisfies, $\max_{g(i)=g(i')} |M_{i,j} - M_{i',j}| \leq c\min_{g(i)\neq g(i')} |M_{i,j} - M_{i',j}|$. If $c < 1/4$, our algorithm remains correct since the search for a single discriminative feature is still meaningful and enables classification. However, if the within-group heterogeneity becomes comparable to or larger than the inter-group differences in some features, our method could fail, and further investigation would be required. Note also that if $c = 1/2$, the problem becomes unidentifiable.

## Acknowledgements

We are grateful to the anonymous reviewers for their careful reading and constructive feedback, which improved the clarity and the rigor of the paper. We also thank Alexandra Carpentier for valuable discussions and insights that contributed to the development of this work.

The work of V. Thuot and N. Verzelen has been partially supported by grant ANR-21-CE23-0035 (ASCAI,ANR). The work of M. Graf has been partially supported by the DFG Forschungsgruppe FOR 5381 "Mathematical Statistics in the Information Age - Statistical Efficiency and Computational Tractability", Project TP 02, and by the DFG on the French-German PRCI ANR ASCAI CA 1488/4-1 "Aktive und Batch-Segmentierung, Clustering und Seriation: Grundlagen der KI".

## Impact Statement

This paper presents work whose goal is to advance the field of Machine Learning. There are many potential societal consequences of our work, none which we feel must be specifically highlighted here.

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

## A. Notation

To ease the reading, we gather the main notation below

- $n$ number of items, $d$ number of features

- $\mu_0 \neq \mu_1 \in \mathbb{R}^d$ feature vectors of the two groups

- $M_{i,\cdot} \in \{\mu_0, \mu_1\}$, $i \in [n]$ feature vector of item $i$

- $M$ matrix with rows $(M_{i,\cdot})$ (with size $n \times d$)

- $g \in \{0, 1\}^n$ true labels (fixing $g(1) = 0$)

- $X \sim \nu_{i,j}$ for $(i, j) \in [n] \times [d]$: $\mathbb{E}[X] = M_{i,j}$ with $X - M_{i,j}$ 1-sub-Gaussian

- $\delta \in (0, 1)$ prescribed probability of error

- $\theta := \frac{\sum_{i=1}^n \mathbb{1}(g(i)=0) \wedge \sum_{i=1}^n \mathbb{1}(g(i)=1)}{n}$ balancedness

- $\Delta := \mu_1 - \mu_0 \neq 0$ gap vector

- $s^* \in \operatorname{argmax}_{s \in [d]} s \cdot \Delta_{(s)}^2$ effective sparcity

Moreover, as we repeatedly compare the entries of $M$ with some row $r_0$ of our choice, we consider a fixed row $r_0$ for the following proofs and define

- $D_{i,j} := M_{i,j} - M_{r_0,j}$ , for $(i, j) \in [n] \times [d]$.

## B. Experiment 4: Varying $\theta$

In this experiment we consider matrices with dimensions $n = d = 1000$. The gap vector $\Delta'$ is defined as

$$\Delta'_j = \begin{cases} 1.5 & \text{if } j \leq 100 \ , \\ 0 & \text{else.} \end{cases}$$

The following simulations are run in $\kappa = 5000$ trials: For different values $\theta_\gamma = 2^\gamma/n$, $\gamma = 0, 1, \ldots, \lfloor \log_2(n) \rfloor - 1$, we run CR and CBC with confidence parameter $\delta = 0.4$ and BanditClustering with $\delta = 0.8$, analogously to experiment 1. The respective budgets are plotted in Figure 4. We also compare our procedure to the $K$-means algorithm by uniformly allocating budgets $T_\iota = \frac{2^\iota - 1}{2^{20} - 1}(10^{10} - 10^6) + 10^6$, $\iota = 0, \ldots, 20$, by looking for each $\theta_\gamma$ for the smallest $T_\iota$ such that at most 0.01 (again, approximately the error rate of BanditCLustering(0.8) for all $\theta_\gamma$) respective Monte Carlo iterations returned an incorrect cluster. We also illustrate these times in Figure 4.

Again, one can see that the budget of BanditClustering is mainly spent on the CBC part of the algorithm. For small values of $\theta$, our algorithm requires less budget than an equally accurate version of the $K$-means algorithm. We can see that the budget of CBC is not effected by $\theta$, while the budget of CR seems to be almost proportional to $1/\theta$. Since $n$, $d$ and $\Delta'$ are fixed, this is in line with the results in Corollary 3.2.

## C. Extension to $K > 2$ clusters

In the main part of this manuscript, we focused on the case of two groups as it serves as an informative baseline for understanding the optimal trade-offs in clustering with bandit feedback. We introduce in this section an algorithm that extends our method to the general case where $K > 2$.

Building on the the ideas of Algorithm 3, we aim to identify a set of representative items $(r_1, \ldots, r_K) \in [n]^K$, with one item from each cluster. Given these representatives, the algorithm learns a discriminative feature for each pair of clusters, enabling perfect clustering in the general case with $K > 2$ groups. The algorithm is described in pseudo-code in Algorithm 5.

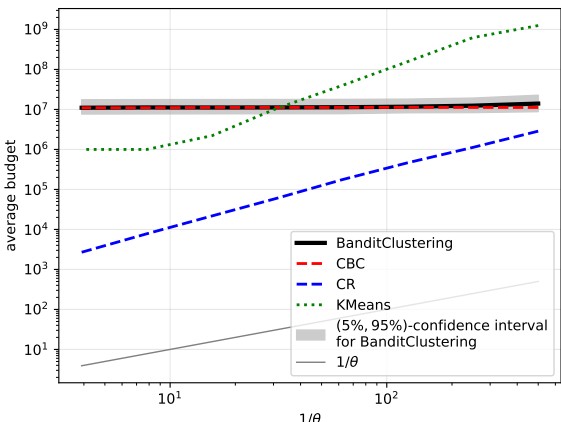

*Figure 4.* Different budgets for Experiment 4, depending on $\theta_\gamma$.

**Setting with $K > 2$ groups.**   In this section, we assume that the items are partitioned into exactly $K \geqslant 2$ clusters, such that items within the same cluster share the same mean-vector. Specifically, there exists $K$ centers in $\mathbb{R}^d$, denoted as $\mu_1, \ldots, \mu_K$ such that, for any $i \in [n]$ then $M_{i,\cdot} \in \{\mu_1, \ldots, \mu_K\}$. We further assume that all the centers $\mu_1, \ldots, \mu_K$ are pairwise distinct. For each $k \in [K]$, we denote as $G_k^* := \{i \in [n] \, ; \, M_{i,\cdot} = \mu_k\}$, and we assume that each cluster $G_k^*$ is non-empty. As before, we consider sub-Gaussian noise and aim to identify the true partition $G^* = G_1^*, \ldots, G_K^*$ in the $\delta$-PAC setting.

Note that the partition $G^* = G_1^* \sqcup \cdots \sqcup G_K^*$ is defined only up to a permutation of the cluster labels.

**Description of Algorithm 5.**   The algorithm uses the subroutines CR (Algorithm 2) and CBC (Algorithm 3), applied to subset of items.

For any $\delta \in (0, 1)$, $r \in [n]$ and $G \subset [n]$, we denote as $\mathrm{CR}(\delta, r; G)$ the call of Algorithm 2 where the search of representatives is restricted to a given set of items $G$ (instead of $[n]$). We recall that $\mathrm{CR}(\delta, r, G)$ is designed, to identify, if it exists, an item $s \in G$ such that $M_{s,\cdot} \neq M_{r,\cdot}$.

For any $\delta \in (0, 1)$, $r \in [n]$, $s \in [n]$ and $G \subset [n]$, we write $\mathrm{CBC}(\delta, r, s; G)$ for the run of CBC restricted to the set of items $G$. As we will prove later, with high probability $1 - \delta$, $\mathrm{CBC}(\delta, r, s; G)$ will output a partition of $G$ into two groups, such that the items with mean $M_{r,\cdot}$ and $M_{s,\cdot}$ are well-separated. When calling $\mathrm{CBC}(\delta, r, s; G)$, the items from the clusters of $r$ and $s$ within $G$ are separated in two distinct sets, while other items might be split anywhere.

The algorithm takes as input the confidence parameter $\delta$ and the number of clusters $K$. It is important to note that, in the $\delta$-PAC setting, the number of clusters $K$ must be known in advance. Indeed, even in the simpler case where there are only two items with means $\mu_1$ and $\mu_2$, there is no finite time testing procedure that can decide between the hypotheses $\mu_1 = \mu_2$ (i.e., $K = 1$) and $\mu_1 \neq \mu_2$ (i.e., $K = 2$) without any prior knowledge on the separation $\mu_1 - \mu_2$.

First, we start with a partition $G^{(1)} = [n]$, where all items are grouped together, and we fix an arbitrary item $r_1 \in G^{(1)}$ as the first representative.

The algorithm proceeds in $K - 1$ epochs, indexed by $e = 1, \ldots, (K - 1)$. In each epoch $e$, the algorithm identifies a new representative $r_{e+1}$ and isolates all items sharing the same mean-vector $M_{r_{e+1},\cdot}$ into a new cluster.

At the beginning of the $e$-th epoch, the algorithm has access to a partition of $[n]$ into $e$ groups $[n] = \cup_{k=1}^e \hat{G}_k^{(e)}$ together with $e$ representatives $r_1, \ldots, r_e$. If all the previous epochs were successful, then the representatives $r_1, \ldots, r_e$ belong to different clusters, and the intermediate cluster $\hat{G}_k^{(e)}$ will contain all items with the same mean-vector as $r_k$. The remaining items (i.e., those from unrepresented clusters) may be mixed in the current partition $\hat{G}_1^{(e)}, \ldots, \hat{G}_e^{(e)}$.

To identify a new representative, the algorithm calls CR with the representative $r_k$ restricted to the group $\hat{G}_k^{(e)}$, for each

$k = 1, \ldots, e$, using confidence level $\delta_e$. These $e$ calls — denoted $\texttt{CR}(\delta_e, r_k; \hat{G}_k^{(e)})$ — are run in parallel. The first returned item from any successful call is selected as the new representative $re + 1$.

Next, for each $k \in [e]$, the algorithm runs $\texttt{CBC}(\delta_e, r_k, r_{e+1}; \hat{G}_k^{(e)})$ to split the group $\hat{G}_k^{(e)}$ based on the new representative. With high probability, $r_k$ and $r_{e+1}$ come from different clusters. In that case, $\texttt{CBC}$ will divide $\hat{G}_k^{(e)}$ into two subgroups: $\hat{G}_k^{(e+1)}$, containing all items with mean $M_{r_k, \cdot}$, and $\hat{R}_k^{(e+1)}$, containing all items with mean $M_{r_{e+1}, \cdot}$. We then define the new group $\hat{G}_{e+1}^{(e+1)} := \bigcup_{k=1}^{e} \hat{R}_k^{(e+1)}$, which should contain all items with mean $M_{r_{e+1}, \cdot}$.

At the end of the $(K-1)$-th epoch, the partition $[n] = \cup_{k=1}^{K} \hat{G}_k^{(K)}$ is the output of the algorithm. With high probability, it should be exact.

---

**Algorithm 5** $K$-$\texttt{BanditClustering}$

---

**Require:** confidence parameter $\delta > 0$, number of clusters $K$
**Ensure:** Clusters $\hat{G}_1, \ldots, \hat{G}_K$
 1: set $\hat{G}_1^{(1)} \leftarrow [n]$
 2: pick a first representative $r_1$ uniformly at random from $[n]$
 3: **for** $1 \leqslant e \leqslant K - 1$ **do**
 4:     set $\delta_e \leftarrow \frac{\delta}{e(K-1)}$
 5:     Run in parallel $\texttt{CR}(\delta_e, r_k; G_k^{(e)})$ for $k \in [r]$ until one new representative $r_{e+1}$ is identified
 6:     **for** $k \in [e]$ **do**
 7:         call $\texttt{CBC}(r_k, r_{e+1}, \delta_e; G_k^{(e)})$ to cluster $G_k^{(e)}$ into two groups $\hat{G}_k^{e+1}$, $\hat{R}_k^{e+1}$ (swith $r_k \in \hat{G}_k^{e+1}$)
 8:     **end for**
 9:     gather $\hat{G}_{e+1}^{e+1} = \cup_{k=1}^{e} \hat{R}_k^{e+1}$
10: **end for**
11: **return** $\hat{G}_1^{(K)} \ldots, \hat{G}_K^{(K)}$ partition of the items

---

**Theorem C.1.** *Let $\nu$ be an environment with $n$ items. Assume that there exists $[n] = \bigsqcup_{k=1}^{K} G_k^*$ a partition of the items into $K$ nonempty and disjoint groups such that all items in $G_k^*$ share the same mean-vector $\mu_k$. For any $k \in [K]$, denote as $\theta_k = \frac{|\{i \in [n] : M_{i,\cdot} = \mu_k\}|}{n}$ as the proportion of items with mean-vector $\mu_k$.*

*Define, for any $0 \neq \Delta \in \mathbb{R}^d$,*

$$\tilde{H}(\Delta) := \min_{s \in [d]} \left[ \left( \frac{d}{s} + n \right) \left( \frac{1}{\Delta_{(s)}^2} + 1 \right) \right] \ .$$

*For $\delta \in (0, 1/e)$, consider Algorithm 5 with entry $\delta$ and $K$.*

*With probability larger than $1 - \delta$, Algorithm 5 returns a partition $\hat{G}$ of $[n]$ equal to $G^*$ (up to labelization of the clusters), with a budget of at most*

$$\tilde{C} \log \left( \frac{1}{\delta} \right) \left[ \sum_{k \in [K]} \max_{k' \neq k} \frac{Kd}{\theta_k \|\mu_k - \mu_{k'}\|^2} + \sum_{k \neq k'} \tilde{H}(\mu_k - \mu_{k'}) \right] \ ,$$

*where there exists a numerical constant $C$, such that $\tilde{C}$ is a logarithmic factor smaller than*

$$C \cdot (\log \log(1/\delta) \vee 1)^4$$
$$\cdot \log(dn)^5 \log(d) (\log_+ \log(d / \min_{k \neq k'} \|\mu_k - \mu_k'\|^2) \vee 1) \ .$$

The proof of Theorem C.1 is postponed to *Appendix G*. The proofs simply exploit the results obtained in the next section of this Appendix about subroutines $\texttt{CR}$ and $\texttt{CBC}$.

**Comments on Algorithm 5 and Theorem C.1** In Algorithm 5, we extend our clustering approach to the general case with $K > 2$ clusters by reducing the problem to a sequence of binary classification tasks. This allows us to reuse the subroutines

CR and CBC, originally designed for the case where there two groups, in a pipeline fashion. As shown in Theorem C.1, this reduction yields a $\delta$-PAC algorithm for the general clustering problem.

The resulting sample complexity scales as $K^2$, since the algorithm performs one binary classification for each pair of clusters. This quadratic dependency is unavoidable in the worst case—for example, when the cluster means $\mu_1, \ldots, \mu_K$ are positioned in such a way that each pair of clusters must be treated independently. However, this approach may be suboptimal in general settings. For instance, if certain features allow simultaneous discrimination among all $K$ clusters, then the sample complexity should not need to scale quadratically in $K$.

In other words, while our extension to $K > 2$ is straightforward and functional, it remains naive in terms of adaptivity. A more refined approach would aim to capture instance-dependent complexity, leveraging the joint geometry of the cluster centers $\mu_1, \ldots, \mu_K$ to potentially reduce the overall budget. Developing such adaptive strategies remains an open and interesting direction for future work.

## D. Analysis of Algorithm 1

We analyze here the performance of CSH.

**Lemma D.1.** *Consider $\delta \in (0,1)$, $s \in [d]$ and $h > 0$ such that $\left|\Delta_{(s)}\right| \geq h$. Consider $I \subset [n]$ and define the relative proportion of items in the second group as $\alpha = \frac{|\{i \in I; \, g(i)=1\}|}{|I|}$ . Consider Algorithm 1– CSH$(r_0, I, L, T)$ –with input $r_0, I, L, T$ such that*

$$L = \left\lceil \log_2 \left( 16 \frac{d}{\alpha s} \log \left( \frac{4 \log(8|I|d)}{\delta} \right) \right) \right\rceil \quad , \tag{6}$$

$$T \geq 516 \frac{L^3 \cdot 2^L}{h^2} \vee 2^{L+1} L \quad . \tag{7}$$

*Then CSH$(r_0, I, L, T)$ outputs a pair $(\hat{i}, \hat{j})$ such that $\left|M_{\hat{i},\hat{j}} - M_{r_0,\hat{j}}\right| \geq h/2$ with probability $\geq 1 - \delta$.*

Remark that for $I = [n]$, then $\alpha \geqslant \theta$. If $I = \{r_1\}$ where $g(r_1) \neq g(r_0)$, then $\alpha = 1$.

Throughout this section, we will prove Lemma D.1 with $I = [n]$. The general result directly follows from the case where $I$ contains all items. To see that, we just have to see that $\alpha$ is equal to the balancedness of the matrix $M|_I$ restricted to the rows in $I$, and we would replace $n$ by $|I|$, and $\theta$ by $\alpha$.

Therefore, we consider Algorithm 1 with input $r_0$, $I = [n]$, $L = \left\lceil \log_2 \left( 16 \frac{d}{\theta s} \log \left( \frac{4 \log(8nd)}{\delta} \right) \right) \right\rceil$ and $T = 516 \frac{L^3 \cdot 2^L}{h^2} \vee 2^{L+1} L$. For simplicity in notation, we also fix $r_0 = 1$ for the proofs.

We recall that we use for the proofs the notation $D_{i,j} := M_{i,j} - M_{r_0,j} = M_{i,j} - M_{1,j}$ for any couple $(i,j) \in [n] \times [d]$ as the gap between the entries of $M$ compared to the mean-vector of the fixed item $r_0 = 1$.

For the following proofs, define $\gamma := h/2L$, and for any halving step $l = l = 0, 1, \ldots, L$, we define $U_l$ as the set of remaining entries $(i,j)$ in $S_l$ such that the gap $|D_{i,j}|$ exceeds $h - l\gamma$

$$U_l := \{(i,j) \in S_l : \, |D_{i,j}| \geq h - l\gamma\} \quad .$$

Lemma D.1 is a direct consequence of following statement:

**Lemma D.2.** *With probability of at least $1 - \delta$, it holds*

$$\frac{|U_l|}{|S_l|} \geq 2^{-L+3} \log \left( \frac{4 \log(8nd)}{\delta} \right) \quad \forall l = 0, 1, \ldots, L \quad .$$

*Proof of Lemma D.1.* The first statement follows by Lemma D.2. Indeed, at the last halving step, $S_L$ contains only one pair of indices $(\hat{i}, \hat{j})$. Lemma D.2 implies that $U_L \subseteq S_L$ is nonempty with probability at least $1 - \delta$, so that $(\hat{i}, \hat{j}) \in U_L$, that is, $D_{\hat{i},\hat{j}} \geqslant h - L\gamma = h/2$. $\qquad\square$

*Proof of Lemma D.2.* We will prove via induction over $l$, that

$$\frac{|U_k|}{|S_k|} \geq 2^{-L+3} \log \left( \frac{4 \log(8nd)}{\delta} \right) \quad \forall k = 0, 1, \ldots, l$$

holds with probability at least

$$1 - (l+1)\left(\frac{\delta}{4\log(8nd)}\right)^2 \ .$$

The statement follows then from

$$
\begin{aligned}
(L+1)\cdot\left(\frac{\delta}{4\log(8nd)}\right)^2 &\leq \left(2\log\left(16\frac{d}{\theta s}\log\left(\frac{4\log(8nd)}{\delta}\right)\right)+1\right)\cdot\left(\frac{\delta}{4\log(8nd)}\right)^2 \\
&\leq 3\log\left(8\frac{d}{\theta s}\right)\cdot\left(\frac{\delta}{4\log(8nd)}\right)^2 + 2\log\log\left(\left(\frac{4\log(8nd)}{\delta}\right)^2\right)\cdot\left(\frac{\delta}{4\log(8nd)}\right)^2 \\
&\leq \frac{3}{4}\frac{\delta^2}{\log(8nd)} + \frac{\delta}{4\log(8nd)} \leq \delta \ ,
\end{aligned}
$$

where we used that $\lceil\log_2(x)\rceil \leq 2\log(x)$ for $x>5$, and the last line is obtained by $8d/\theta s \leq 8nd$ and $2x\cdot\log\log(1/x) \leq \sqrt{x}$ for $x\in(0,1)$.

**The base case $l=0$** The initial set $S_0 = \{(i_1,j_1),\ldots,(i_{2^L},j_{2^L})\}$ is constructed by picking $2^L$ entries uniformly at random (with replacement) from $[n]\times[d]$), as described in Line 1 in Algorithm 1. In the context of Lemma D.2, the parameter $\alpha$ reduces to $\theta$, since we consider $I=[n]$ for the proof, and $s,h$ are such that $|\Delta_{(s)}\geqslant h$. Consequently, the matrix $M$ contains at least $\theta n\cdot s$ entries such that $|D_{i,j}|\geq h$. Then, the random variables

$$X_t^{(0)} := \mathbb{1}\left(|D_{i_t,j_t}|\geq h\right), \quad t=1,\ldots,2^L$$

are i.i.d. Bernoulli random variables with $\mathbb{P}(X_t^{(0)}=1)\geq\theta\frac{s}{d}$. In particular, we have

$$\mathbb{E}\left[\sum_{t=0}^{2^L}X_t^{(0)}\right] = 2^L\theta\frac{s}{d} \geq 16\log\left(\frac{4\log(8nd)}{\delta}\right) \ .$$

Applying the second inequality in Lemma I.1 (a standard Chernoff bound for Bernoulli distributions), we obtain:

$$\mathbb{P}\left(\sum_{t=0}^{2^L}X_t^{(0)}\leq 8\log\left(\frac{4\log(8nd)}{\delta}\right)\right) \leq \mathbb{P}\left(\sum_{t=0}^{2^L}X_t^{(0)}\leq\frac{1}{2}\mu_{(0)}\right) \leq \exp\left(-\frac{\mu_{(0)}}{8}\right) \leq \left(\frac{\delta}{4\log(8nd)}\right)^2 \ .$$

So we have

$$|U_0| = \sum_{t=0}^{2^L}X_t^{(0)} > 8\log\left(\frac{4\log(8nd)}{\delta}\right) = |S_0|2^{-L+3}\log\left(\frac{4\log(8nd)}{\delta}\right)$$

with probability at least $1-\left(\frac{\delta}{4\log(8nd)}\right)^2$.

**Induction step: from $l$ to $l+1$** Consider the event $\xi_l$, defined as

$$\frac{|U_k|}{|S_k|}\geq 2^{-L+3}\log\left(\frac{4\log(8nd)}{\delta}\right) \quad \forall\, k=0,1,\ldots,l \ .$$

We want to show

$$\mathbb{P}(\xi_l)\geq 1-(l+1)\left(\frac{\delta}{4\log(8nd)}\right)^2 \quad\Rightarrow\quad \mathbb{P}(\xi_l)\geq 1-(l+2)\left(\frac{\delta}{4\log(8nd)}\right)^2$$

Note that $\xi_{l+1}\subseteq\xi_l$, so showing

$$\mathbb{P}(\xi_{l+1}\mid\xi_l)\geq 1-\left(\frac{\delta}{4\log(8nd)}\right)^2$$

suffices to conclude

$$\mathbb{P}\left(\xi_{l+1}\right) = \mathbb{P}\left(\xi_l\right) \cdot \mathbb{P}\left(\xi_{l+1} \mid \xi_l\right) \geq \left(1 - (l+1)\left(\frac{\delta}{4\log(8nd)}\right)^2\right)\left(1 - \left(\frac{\delta}{4\log(8nd)}\right)^2\right)$$

$$\geq 1 - (l+2)\left(\frac{\delta}{4\log(8nd)}\right)^2 \ .$$

When we condition on the event $\xi_l$, this implies the condition

$$|U_l| \geq 2^{-L+3}|S_l|\log\left(\frac{4\log(8nd)}{\delta}\right) \ .$$

Recall that in line 5 of Algorithm 1, we first sample

$$X_{1,j}^{(1)}, \dots, X_{1,j}^{(\tau_{l+1})} \sim^{\text{i.i.d.}} \nu_{1,j} \ , X_{i,j}^{(1)}, \dots, X_{i,j}^{(\tau_{l+1})} \sim^{\text{i.i.d.}} \nu_{i,j}$$

for each $(i,j) \in S_l$ and store

$$\hat{D}_{i,j} = \frac{1}{\tau_{l+1}}\sum_{u=1}^{\tau_{l+1}} X_{i,j}^{(u)} - X_{1,j}^{(u)} \ .$$

Since we assumed that $X_{i,j}^{(u)} - M_{i,j} \in SG(1)$ and $X_{1,j}^{(u)} - M_{1,j} \in SG(1)$, this implies

$$\sum_{u=1}^{\tau_{l+1}} \left(X_{i,j}^{(u)} - X_{1,j}^{(u)} - D_{i,j}\right) \in SG(2\tau_{l+1})$$

and we obtain

$$\mathbb{P}\left(\hat{D}_{i,j} - D_{i,j} \geq \gamma/2\right) = \left(\sum_{u=1}^{\tau_{l+1}}\left(X_{i,j}^{(u)} - X_{1,j}^{(u)} - D_{i,j}\right) \geq \tau_{l+1}\gamma/2\right) \leq \exp\left(-\frac{\tau_{l+1}\gamma^2}{16}\right) =: p_l \ , \qquad (8)$$

and likewise

$$\mathbb{P}\left(D_{i,j} - \hat{D}_{i,j} \geq \gamma/2\right) \leq p_l \ ,$$

For $(i,j) \in U_l$, this implies that

$$\mathbb{P}\left(\left|\hat{D}_{i,j}\right| \leq h - (l+1/2)\gamma\right) \leq p_l \ .$$

So we can construct i.i.d. Bernoulli random variables $B_{i,j}$ with

$$\mathbb{P}(B_{i,j} = 1) = p_l$$

and

$$\left|\hat{D}_{i,j}\right| \leq h - (l+1/2)\gamma \quad \Rightarrow \quad B_{i,j} = 1$$

for $(i,j) \in U_l$. By Lemma I.1 it follows (by letting $\kappa = \frac{1}{2p_l} - 1$)

$$\mathbb{P}\left(\sum_{(i,j)\in U_l} B_{i,j} \geq |U_l|/2 \mid |U_l| = \eta\right) \leq \exp\left(\kappa p_l \eta - (1+\kappa)p_l n \log(1+\kappa)\right)$$

$$\leq \exp\left(\eta\left(1/2 - p_l - \frac{\log(1/p_l) - \log(2)}{2}\right)\right)$$

$$\leq \exp\left(\eta\left(1/2 - \exp(-\tau_{l+1}\gamma^2/16) - \frac{\tau_{l+1}\gamma^2/16 - \log(2)}{2}\right)\right)$$

$$\leq \exp(-2^{l-2}\eta) \ .$$

The last inequality follows, since we assumed

$$T \geq 516 \cdot 2^L \cdot L^3/h^2 \vee 2^{L+1}L \geq 128 \cdot 2^L \cdot L/\gamma^2 \vee 2^{L+1}L \ ,$$

such that

$$\tau_{l+1} = \left\lfloor \frac{T}{2^{L-l+1}L} \right\rfloor \geq \left\lfloor 32\frac{2^{l+1}}{\gamma^2} \vee 2^{l-1} \right\rfloor \geq 16\frac{2^{l+1}}{\gamma^2}$$

(by $\lfloor x \rfloor \geq x/2$ for $x \geq 1$), from where we can conclude

$$1/2 - \exp(-\tau_{l+1}\gamma^2/16) - \frac{\tau_{l+1}\gamma^2/16 - \log(2)}{2} \leq -2^{l-2}$$

for $l \geq 0$. For

$$\eta \geq 2^{-L+3}|S_l|\log\left(\frac{4\log(8nd)}{\delta}\right) = 2^{-l+2}\log\left(\left(\frac{4\log(8nd)}{\delta}\right)^2\right) \ ,$$

this implies

$$\mathbb{P}\left(\sum_{(i,j)\in U_l} B_{i,j} \geq |U_l|/2 \mid |U_l| = \eta\right) \leq \left(\frac{\delta}{4\log(8nd)}\right)^2$$

and therefore

$$\mathbb{P}\left(\sum_{(i,j)\in U_l} B_{i,j} \geq |U_l|/2 \mid \xi_l\right) \leq \left(\frac{\delta}{4\log(8nd)}\right)^2 \tag{9}$$

Next, define

$$V_l := \{(i,j) \in S_l : |D_{i,j}| < h - (l+1)\gamma\} \ .$$

Note that $S_{l+1} \setminus U_{l+1} \subseteq V_l$. So if

$$|V_l| < 2^{-L+3}|S_l|\log\left(\frac{4\log(8nd)}{\delta}\right) \ ,$$

this implies

$$\begin{aligned}
|U_{l+1}| &= |S_{l+1}| - |S_{l+1} \setminus U_{l+1}| \\
&\geq |S_{l+1}| - |V_l| \\
&> |S_{l+1}| - 2^{-L+3}|S_l|\log\left(\frac{4\log(8nd)}{\delta}\right) \\
&= \left(1 - 2^{-L+2}\right)|S_{l+1}|\log\left(\frac{4\log(8nd)}{\delta}\right) \\
&\geq 2^{-L+3}|S_{l+1}|\log\left(\frac{4\log(8nd)}{\delta}\right) \ ,
\end{aligned} \tag{10}$$

since we have $L \geq \log_2(16) = 4$. Therefore, consider the nontrivial case

$$|V_l| \geq 2^{-L+3}|S_l|\log\left(\frac{4\log(8nd)}{\delta}\right) \ ,$$

Like before, we have from (8) that

$$\mathbb{P}\left(\left|\hat{D}_{i,j}\right| \geq h - (l+1/2)\gamma\right) \leq p_l$$

for $(i,j) \in V_l$. Note that we can again define Bernoulli random variables $C_{i,j}$ with

$$\mathbb{P}(C_{i,j} = 1) = p_l$$

and

$$|D_{i,j}| \geq h - (l + 1/2)\gamma \quad \Rightarrow \quad C_{i,j} = 1$$

for all $(i, j) \in V_l$. We can again show that conditional $|V_l| = \eta$ with

$$\eta \geq 2^{-L+3}|S_l|\log\left(\frac{4\log(8nd)}{\delta}\right) ,$$

it holds

$$\sum_{(i,j)\in V_l} C_{i,j} \geq \eta/2 \tag{11}$$

with probability $1 - \left(\frac{\delta}{4\log(8nd)}\right)^2$.

Now if $\overline{\Delta}$ is the median of the $\hat{D}_{i,j}$, $(i, j) \in S_l$, it is either $\overline{\Delta} < h - (l + 1/2)\gamma$ or $\overline{\Delta} \geq h - (l + 1/2)\gamma$. In the case $\overline{\Delta} < h - (l + 1/2)\gamma$, the bound (9) tells us that $U_{l+1}$ contains at least half of the indices of $U_l$, in other words,

$$\frac{|U_{l+1}|}{|S_{l+1}|} \geq \frac{|U_l|}{2|S_{l+1}|} = \frac{|U_l|}{|S_l|} ,$$

with probability at least $1 - \delta$. In the case $\overline{\Delta} \geq h - (l + 1/2)\gamma$, we either directly conclude the induction step from (10), or we know from (11) that the number of $(i, j) \in S_{l+1}$ with $|D_{i,j}| \leq h - (l + 1)\gamma$ is less than half the number of arms in $V_l$, and in particular,

$$\frac{|U_{l+1}|}{|S_{l+1}|} \geq 1 - \frac{|V_l|}{2|S_{l+1}|} = 1 - \frac{|V_l|}{|S_l|} = \frac{|U_l|}{|S_l|} ,$$

with probability at least $1 - \left(\frac{\delta}{4\log(8nd)}\right)^2$. Combining both cases yields the claim. $\qquad \square$

## E. Analysis of Algorithm 2

Using the results of Lemma D.1, we can now determine theoretical guarantees for Algorithm 2, $\text{CR}(\delta, r_0)$.

We present the individual guarantees offered by Algorithm 2 in the following proposition.

**Proposition E.1.** *Let $\delta \in (0, 1/e)$. Then, with probability larger than $1 - \delta$, Algorithm 2– $\text{CR}(\delta, r_0)$ –returns an index $r_1$, such that it holds that $M_{r_1,\cdot} \neq M_{r_0,\cdot}$, and moreover the total budget is upper bounded by*

$$\tilde{C} \cdot \log\left(\frac{1}{\delta}\right) \cdot \frac{d}{\theta}\left(\frac{1}{\|\Delta\|^2} + \frac{1}{s^*}\right) ,$$

*where $\tilde{C}$ is a logarithmic factor smaller than*

$$C \cdot (\log\log(1/\delta) \vee 1)^4 \cdot \log(dn)^5 \log(d)(\log_+ \log(1/\Delta_{(s^*)}^2) \vee 1) ,$$

*with a numerical constant $C > 0$ and $\log_+(x) := \log(x \vee 1)$ for $x \in \mathbb{R}$.*

*Proof of Proposition E.1.* Consider $s \in [d]$ and $k \geq 1$ minimal, such that for

$$L = \left\lceil \log_2\left(16\frac{d}{\theta s}\log\left(\frac{16\log(8nd)}{\delta}\right)\right)\right\rceil$$

it holds

$$516\frac{L^3 2^L}{\Delta_{(s)}^2} \vee 2^{L+1}L \leq 2^k \tag{12}$$

and

$$3714 \cdot \frac{\log(1/\delta) + \log_+ \log\left(1/\Delta_{(s)}^2\right)}{\Delta_{(s)}^2} \vee 2 < 2^k . \tag{13}$$

The proof consists of three parts:

1. $\text{CSH}(r_0, [n], L, 2^k)$ returns $(r_1, j)$ with $|D_{r_1,j}| \geq |\Delta_{(s)}|/2$, with probability at least $1 - \delta/2$,

2. for $|D_{r_1,j}| \geq |\Delta_{(s)}|/2$, if we sample

$$X_{r_0,j}^{(1)}, \ldots, X_{r_0,j}^{(2^k)} \sim^{\text{i.i.d.}} \nu_{r_0,j} \quad \text{and} \quad X_{r_1,j}^{(1)}, \ldots, X_{r_1,j}^{(2^k)} \sim^{\text{i.i.d.}} \nu_{r_1,j} \ ,$$

it holds with probability of at least $1 - \frac{\delta}{0.3k^3}$ that

$$\left| \frac{1}{2^k} \sum_{t=1}^{2^k} X_{r_1,j}^{(t)} - X_{r_0,j}^{(t)} \right| > \sqrt{\frac{4}{2^k} \log\left(\frac{k^3}{0.15\delta}\right)} \ ,$$

3. for any $k' \geq 1$, if $D_{i,j'} = 0$ and we sample

$$X_{r_0,j'}^{(1)}, \ldots, X_{r_0,j'}^{(2^{k'})} \sim^{\text{i.i.d.}} \nu_{r_0,j'} \quad \text{and} \quad X_{i,j'}^{(1)}, \ldots, X_{i,j'}^{(2^{k'})} \sim^{\text{i.i.d.}} \nu_{i,j'} \ ,$$

it holds with probability of at least $1 - \frac{\delta}{0.3k'^3}$ that

$$\left| \frac{1}{2^{k'}} \sum_{t=1}^{2^{k'}} X_{i,j'}^{(t)} - X_{r_0,j'}^{(t)} \right| \leq \sqrt{\frac{4}{2^{k'}} \log\left(\frac{k'^3}{0.15\delta}\right)} \ .$$

From point 1 and 2 one can conclude that Algorithm 2 terminates in the $L^{\text{th}}$ step of the $k^{\text{th}}$ iteration at the latest with probability at least $1 - \delta/2 - \delta/0.3k^3$. If it has not terminated before, by point 1, we obtain in line 4 $(r_1, j)$ with $|D_{r_1,j}| \geq |\Delta_{(s)}|/2$ and by point 2, that for such $(r_1, j)$, the algorithm terminates in line 6, returning $r_1$.

If the algorithm terminates for some $k' < k$ or in the $k^{\text{th}}$ round, but for some other $L'$, by line 6 this means that the algorithm returns some $i$ such that for some $j'$ it holds

$$\left| \frac{1}{2^{k'}} \sum_{t=1}^{2^{k'}} X_{i,j'}^{(t)} - X_{r_0,j'}^{(t)} \right| > \sqrt{\frac{4}{2^{k'}} \log\left(\frac{k'^3}{0.15\delta}\right)} \ .$$

So point 3 implies for each iteration $k'$ and each $L'$ we iterate over, that we do not return an index $i$ with $M_{i,\cdot} = M_{r_0,\cdot}$, with probability at least $1 - \delta/0.3k^3$.

So by the union bound, Algorithm 2 returns $r_1$ with $M_{r_1,\ldots} \neq M_{r_0,\ldots}$ with probability at least

$$1 - \delta/2 - \sum_{k' \leq k} \sum_{\substack{1 \leq L \leq L_{\max} \\ L2^L \leq 2^{k'}}} \delta/(0.3k'^3) \geq 1 - \delta/2 - \delta \cdot 0.3 \sum_{k \geq 1} \frac{1}{k^2} \geq 1 - \delta/2 - 0.3\frac{\pi^2}{6}\delta > 1 - \delta \ .$$

So we are left with proving the three points.

**Proof of 1**  By Lemma D.1 and inequality (12), calling $\text{CSH}(r_0, [n], L, 2^k)$ in line 4 of Algorithm 2 yields a pair $(r_1, j)$ with $|D_{r_1,j}| \geq |\Delta_{(s)}|/2$ with probability at least $1 - \delta/2$.

**Proof of 2**  Note that for

$$\hat{D}_{r_1,j} := \frac{1}{2^k} \sum_{t=1}^{2^k} X_{r_1,j}^{(t)} - X_{r_0,j}^{(t)} \ , \tag{14}$$

by an application of Hoeffding's inequality we know

$$\hat{D}_{r_1,j} \in \left[ D_{r_1,j} - \sqrt{\frac{4}{2^k} \log\left(\frac{k^3}{0.15\delta}\right)}, D_{r_1,j} + \sqrt{\frac{4}{2^k} \log\left(\frac{k^3}{0.15\delta}\right)} \right] \tag{15}$$

with a probability of at least $1 - 0.3\delta/k^3$. Note that from inequality (13) we know by the monotonicity of $\log\log(x)/x$ for $x \geq e^2$ that

$$\frac{16}{2^k}\log\left(\frac{k^3}{0.15\delta}\right) = \frac{16}{2^k}\left(\log(1/\delta) + 3\log\log(2^k) + 3\log\left(\frac{1}{\log(2)}\right) + \log(20/3)\right) \leq 80\frac{\log(1/\delta) + \log\log\left(2^k\right)}{2^k} \ .$$

We want to prove the bound

$$\frac{16}{2^k}\log\left(\frac{k^3}{0.15\delta}\right) \leq \Delta_{(s)}^2/4 \ . \tag{16}$$

Let us first consider the case $\Delta_{(s)}^2 \geq 1/e$. We can bound

$$80\frac{\log(1/\delta) + \log\log(2^k)}{2^k} \leq 80\frac{\log(1/\delta) + \log(2^k)}{2^k}$$

$$= 80\frac{\log(1/\delta) + \log(2^k\Delta_{(s)}^2) - \log(\Delta_{(s)}^2)}{2^k\Delta_{(s)}^2}\Delta_{(s)}^2$$

$$\leq 80\frac{2\log(1/\delta) + \log\log(2^k\Delta_{(s)}^2)}{2^k\Delta_{(s)}^2} \ .$$

From inequality (12), we know

$$2^k\Delta_{(s)}^2 \geq 3714\log(1/\delta) \ ,$$

and we can therefore use that $x \mapsto \log(x)/x$ is decreasing for $x \geq e$ to obtain

$$80\frac{\log(1/\delta) + \log\log(2^k)}{2^k} \leq 80\frac{2\log(1/\delta) + \log(3714\log(1/\delta))}{3714\log(1/\delta)}\Delta_{(s)}^2$$

$$\leq 80\frac{3 + \log(3714)}{3714}\Delta_{(s)}^2 \leq \Delta_{(s)}^2/4 \ .$$

Next, consider the case $\Delta_{(s)}^2 \leq 1/e$. Then we know from inequality(13) that

$$2^k \geq 3714\frac{\log(1/\delta) + \log\log(1/\Delta_{(s)}^2)}{\Delta_{(s)}^2} \ .$$

Because $x \mapsto \log\log(x)/x$ is decreasing for $x \geq e^2$, we can bound

$$80\frac{\log(1/\delta) + \log\log(2^k)}{2^k} \leq 80\frac{\log(1/\delta) + \log\log\left(3714\frac{\log(1/\delta) + \log\log(1/\Delta_{(s)}^2)}{\Delta_{(s)}^2}\right)}{3714\left(\log(1/\delta) + \log\log\left(1/\Delta_{(s)}^2\right)\right)}\Delta_{(s)}^2 \ .$$

For $a, b \geq e$ it holds $\log\log(ab) \leq \log(2) + \log\log(a) + \log\log(b)$, so we can bound

$$\log\log\left(3714\frac{\log(1/\delta) + \log\log(1/\Delta_{(s)}^2)}{\Delta_{(s)}^2}\right)$$

$$\leq \log(2) + \log\log(1/\Delta_{(s)}^2) + \log\log\left(3714\left(\log(1/\delta) + \log\log\left(1/\Delta_{(s)}^2\right)\right)\right)$$

$$\leq \log(2 \cdot 3714) + 2\log\log\left(1/\Delta_{(s)}^2\right) + \log(1/\delta) \ .$$

This allows us to bound

$$80\frac{\log(1/\delta) + \log\log(2^k)}{2^k} \leq 80\frac{(2 + \log(2 \cdot 3714))\log(1/\delta) + 3\log\log\left(1/\Delta_{(s)}^2\right)}{3714\left(\log(1/\delta) + \log\log\left(1/\Delta_{(s)}^2\right)\right)}\Delta_{(s)^2}$$

$$\leq \frac{80(2 + \log(2 \cdot 3714))}{3714}\Delta_{(s)}^2 \leq \Delta_{(s)}^2/4 \ .$$

For $|D_{r_1,j}| \geq |\Delta_{(s)}|/2$, we have with high probability according to (15) that

$$\left| \hat{D}_{r_1,j} \right| \geq \sqrt{\frac{4}{2^k} \log\left( \frac{k^3}{0.15\delta} \right)} \enspace.$$

**Proof of 3**    Analogously to (14) and (15) we can use Hoeffding's inequality to show that for

$$\hat{D}_{i,j} := \frac{1}{2^{k'}} \sum_{t=1}^{2^{k'}} X_{ij}^{(t)} - X_{1j}^{(t)}$$

it holds

$$\left| \hat{D}_{i,j} \right| \leq \sqrt{\frac{4}{2^{k'}} \log\left( \frac{k'^3}{0.15\delta} \right)}$$

with probability at least $1 - \frac{\delta}{0.3k'^3}$.

**Bounding the budget:**    First, we can bound

$$L_{\max} \leq 2 \log\left( 16nd \log\left( \frac{16 \log(8nd)}{\delta} \right) \right)$$

$$\leq 10 \log\left( nd \log\left( \frac{16 \log(8nd)}{\delta} \right) \right)$$

$$\leq 10 \left( \log(nd) + \log\log\left( \frac{16 \log(8nd)}{\delta} \right) \right)$$

$$\leq 10 \left( \log(nd) + \log\left( (nd)^4 + \log(1/\delta) \right) \right)$$

$$\leq 70 \left( \log(nd) + \log\log(1/\delta) \right) \enspace.$$

At the same time, we have that

$$2^L \leq 32 \frac{d}{\theta s} \log\left( \frac{16 \log(8nd)}{\delta} \right)$$

$$\leq 32 \frac{d}{\theta s} \left( \log\log(nd) + \log(64/\delta) \right)$$

$$\leq 192 \frac{d}{\theta s} \left( \log\log(nd) + \log(1/\delta) \right) \enspace.$$

So, if we define $C := 156 \cdot 70^3 \cdot 192$ and $k^*$ the minimal $k \geq 1$ such that

$$2^{k+1} \geq C \min_{s\in[d]} \left( \frac{(\log(nd) + \log\log(1/\delta))^3 (\log\log(nd) + \log(1/\delta))d}{\theta s \Delta_{(s)}^2} \right.$$

$$\left. + \frac{d(\log(nd) + \log\log(1/\delta))(\log\log(nd) + \log(1/\delta))}{\theta s} + \frac{d}{\theta s} \frac{\log(1/\delta) + \log_+ \log(1/\Delta_{(s)}^2)}{\Delta_{(s)}^2} + \right) \enspace,$$

we can see that by (12) and (13) Algorithm 2 terminates and returns $r_1$ such that $M_{r_1,\ldots} \neq M_{r_0,\ldots}$ with a probability of at least $1 - \delta$. Moreover, on this event of high probability, the algorithm terminates after at most

$$\sum_{k=1}^{k^*} \sum_{1 \leq L \leq L_{\max}:\ L\cdot 2^L \leq 2^{k+1}} 2 \cdot 2^{k+1} \leq 8 L_{\max} 2^{k^*}$$

$$\leq C' \cdot L_{\max} \min_{s\in[d]} \left( \frac{(\log(nd) + \log\log(1/\delta))^3 (\log\log(nd) + \log(1/\delta))d}{\theta s \Delta_{(s)}^2} \right.$$

$$\left. + \frac{d(\log(nd) + \log\log(1/\delta))(\log\log(nd) + \log(1/\delta))}{\theta s} + \frac{d}{\theta s} \frac{\log(1/\delta) + \log_+ \log(1/\Delta_{(s)}^2)}{\Delta_{(s)}^2} \right) \enspace,$$

by minimality of $k^*$, where $C' > 0$ is some numerical constant that might change. To obtain the claimed upper bound, note that by (2) we know $1/s^* \Delta^2_{(s^*)} \leq \log(2d)/\|\Delta\|^2_2$ and therefore

$$C' L_{\max} \min_{s \in [d]} \left( \frac{(\log(nd) + \log\log(1/\delta))^3 (\log\log(nd) + \log(1/\delta))d}{\theta s \Delta^2_{(s)}} \right.$$

$$+ \frac{d(\log(nd) + \log\log(1/\delta))(\log\log(nd) + \log(1/\delta))}{\theta s} + \frac{d}{\theta s} \frac{\log(1/\delta) + \log_+ \log(1/\Delta^2_{(s)})}{\Delta^2_{(s)}} + \left. \right)$$

$$\leq C''(\log(nd) + \log\log(1/\delta))^4 (\log(nd) + \log(1/\delta)(\log_+ \log(1/\Delta^2_{s^*}) \vee 1) \frac{d}{\theta} \left( \frac{1}{\|\Delta\|^2} + \frac{1}{s^*} \right) \ ,$$

where $C'' > 0$ is some numerical constant. Reassembling the logarithmic terms yields the claim. $\qquad \square$

## F. Analysis of Algorithm 3

Now, we prove the correctness and we upper bound the budget of Algorithm 3.

**Proposition F.1.** *Let $\delta \in (0, 1/e)$, let $r_1 \in [n]$ such that $M_{r_1,\cdot} \neq M_{r_0,\cdot}$. Then Algorithm 3– $\mathrm{CBC}(\delta, r_0, r_1)$ –returns $\hat{g} = g$ (fixing arbitrary $g(r_0) = 0$), with probability at least $1 - \delta$, with a budget of at most*

$$\tilde{C} \cdot \log(1/\delta) \cdot \min_{s \in [d]} \left[ \left( \frac{d}{s} + n \right) \left( \frac{1}{\Delta^2_{(s)}} + 1 \right) \right] \ ,$$

*where $\tilde{C}$ is a logarithmic factor smaller than*

$$C \cdot (\log\log(1/\delta) \vee 1)^4 \cdot \log(d)^5 \cdot \log_+ \log\left(1/\Delta^2_{(\tilde{s})}\right) \ ,$$

*with a numerical constant $C > 0$, and $\tilde{s} = \lceil d/n \rceil \wedge |\{j \in [d] , \Delta_j \neq 0\}|$.*

The proof of Proposition F.1 does not differ much from the proof of Proposition E.1. Again, we have to bound the time where CSH returns an index pair for which the stopping condition is fulfilled with high probability. The main difference is, that we also need a guarantee for correct clustering using these indices, which also leads to a change of the stopping rule.

*Proof.* Consider $s \in [d]$ and $k \in \mathbb{N}$, $k > \log_2(n)$ minimal, such that for

$$L = \left\lceil \log_2 \left( 16 \frac{d}{s} \log \left( \frac{16 \log(8d)}{\delta} \right) \right) \right\rceil$$

it holds

$$516 \frac{L^3 2^L}{\Delta^2_{(s)}} \vee 2^{L+1} L \leq 2^k \tag{17}$$

and

$$34423 \cdot \frac{\left( \log(1/\delta) + \log_+ \log\left(1/\Delta^2_{(s)}\right) + \log n \right) \cdot n}{\Delta^2_{(s)}} \vee 2n \leq 2^k \ . \tag{18}$$

The proof relies on the two following facts:

1. $\mathrm{CSH}(r_0, [n], L, 2^k)$ returns $(r_1, j)$ with $|D_{r_1,j}| \geq |\Delta_{(s)}|/2$, with probability at least $1 - \delta/2$,

2. we have that jointly for all iterations $k' \geq 1$ and $1 \leq L \leq \tilde{L}_{\max}$ with $2^L L \leq 2^{k+1}$, for some $j' \in [d]$ (chosen each time in line 4) and all $1 < i \leq n$, when we draw

$$X^{(1)}_{r_0,j'}, \dots, X^{(\lfloor 2^{k'}/n \rfloor)}_{r_0,j'} \sim^{\text{i.i.d.}} \nu_{r_0,j'} \quad \text{and} \quad X^{(1)}_{i,j'}, \dots, X^{(\lfloor 2^{k'}/n \rfloor)}_{i,j'} \sim^{\text{i.i.d.}} \nu_{i,j'}$$

holds

$$\left| \sum_{t=1}^{\lfloor 2^{k'}/n \rfloor} \left( X_{i,j'}^{(t)} - X_{1,j'}^{(t)} - D_{i,j'} \right) \right| \leq \sqrt{4 \cdot \lfloor 2^{k'}/n \rfloor \log\left( \frac{nk'^3}{0.15\delta} \right)} \ ,$$

uniformly with probability at least $1 - \delta/2$.

Point 1 is a direct consequence of Lemma D.1 and (17). Point 2 follows directly from Hoeffding's inequality and a union bound over all $k \geq 1$, $L \leq L_{\max}$ such that $L2^L \leq 2^{k+1}$ and $i \in [n]$. Indeed, each inequality for itself holds with probability at least $1 - 0.3\delta/nk'^3$, so the intersection must hold with a probability of at least

$$1 - \sum_{k' \geq 1} \sum_{\substack{1 \leq L \leq L_{\max} \\ L2^L \leq 2^{k'+1}}} \sum_{i=2}^{n} 0.3\delta/nk'^3 \geq 1 - 0.3\delta \sum_{k' \geq 1} \frac{1}{k'^2} \geq 1 - \delta/2 \ .$$

We will prove that Algorithm 3 terminates at the latest in the $L^{\text{th}}$ round of the $k^{\text{th}}$ iteration and clusters correctly with probability at least $1 - \delta$, namely on the intersection of the high probability events of point 1 and 2 which we will call $\xi_{\text{cbc}}$.

**Algorithm 3 terminates at the latest in the $k^{\text{th}}$ iteration** Assume we are on $\xi_{\text{cbc}}$. By point 1, we know that at round $L$ of iteration $k$ it holds $|D_{r_1,j}| \geq |\Delta_{(s)}|/2$ for $j$ obtained in line 4. We want to prove

$$\frac{64}{\left\lfloor \frac{2^k}{n} \right\rfloor} \log\left( \frac{nk^3}{0.15\delta} \right) \leq \Delta_{(s)}^2/4 \ . \tag{19}$$

Note that by (18), it holds

$$\frac{64}{\left\lfloor \frac{2^k}{n} \right\rfloor} \log\left( \frac{nk^3}{0.15\delta} \right) \leq \frac{128n}{2^k} \left( \log n + 3\log\log 2^k + \log(20/3) + \log(1/\delta) \right) \leq 640n \frac{\log(1/\delta) + \log n + \log\log 2^k}{2^k} \ .$$

Again, consider first the case $\Delta_{(s)}^2 \geq 1/e$. In this case, we know from (18) that

$$34423 \left( \log(1/\delta) + \log n \right) \cdot n \leq 2^k \Delta_{(s)}^2 \ .$$

We can use that $x \mapsto \log(x)/x$ is decreasing for $x \geq e$ and obtain

$$
\begin{aligned}
640n \frac{\log(1/\delta) + \log n + \log\log(2^k)}{2^k} &\leq 640n \frac{\log(1/\delta) + \log n + \log(2^k)}{2^k} \\
&= 640n \frac{\log(1/\delta) + \log n + \log(2^k \Delta_{(s)}^2) - \log(\Delta_{(s)}^2)}{2^k \Delta_{(s)}^2} \Delta_{(s)}^2 \\
&\leq 640n \frac{2\log(1/\delta) + \log n + \log(2^k \Delta_{(s)}^2)}{2^k \Delta_{(s)}^2} \Delta_{(s)}^2 \\
&\leq \frac{640}{34423} \cdot \frac{2\log(1/\delta) + \log n + \log\left( 34423n(\log(1/\delta) + \log n) \right)}{\log(1/\delta) + \log n} \Delta_{(s)}^2 \\
&\leq \frac{640}{34423} \cdot \frac{2\log(1/\delta) + 2\log n + \log 34423 + \log(\log(1/\delta) + \log n)}{\log(1/\delta) + \log n} \Delta_{(s)}^2 \\
&\leq \frac{640 \cdot (3 + \log(34423))}{34423} \Delta_{(s)}^2 \leq \Delta_{(s)}^2/4 \ .
\end{aligned}
$$

This proves (19) in the case $\Delta_{(s)}^2 \geq 1/e$.

Consider $\Delta_{(s)}^2 \geq 1/e$. Then, by (18), we know

$$2^k \geq 34423n \frac{\log(1/\delta) + \log n + \log\log\left(1/\Delta_{(s)}^2\right)}{\Delta_{(s)}^2} \quad .$$

We can apply that $x \mapsto \log\log(x)/x$ is decreasing for $x \geq e^2$ and obtain

$$640n \frac{\log(1/\delta) + \log n + \log\log 2^k}{2^k} \leq \frac{640}{34423} \frac{\log(1/\delta) + \log n + \log\log\left(34423n \frac{\log(1/\delta) + \log n + \log\log\left(1/\Delta_{(s)}^2\right)}{\Delta_{(s)}^2}\right)}{\log(1/\delta) + \log n + \log\log\left(1/\Delta_{(s)}^2\right)} \Delta_{(s)}^2 \quad .$$

Note, that

$$\log\log\left(34423n \frac{\log(1/\delta) + \log n + \log\log\left(1/\Delta_{(s)}^2\right)}{\Delta_{(s)}^2}\right)$$

$$\leq \log(2) + \log\log\left(1/\Delta_{(s)}^2\right) + \log\log\left(34423n\left(\log(1/\delta) + \log n + \log\log\left(1/\Delta_{(s)}^2\right)\right)\right)$$

$$\leq \log(2 \cdot 34423) + \log\log\left(1/\Delta_{(s)}^2\right) + \log(n) + \log\left(\log(1/\delta) + \log n + \log\log\left(1/\Delta_{(s)}^2\right)\right)$$

$$\leq (\log(2 \cdot 34423) + 1)\log(1/\delta) + 2\log\log\left(1/\Delta_{(s)}^2\right) + 2\log(n) \quad ,$$

where we used $\log\log(a \cdot b) \leq \log(2) + \log\log(a) + \log\log(b)$ for $a, b \geq e$. Thus, it holds

$$640n \frac{\log(1/\delta) + \log n + \log\log(2^k)}{2^k} \leq \frac{640}{34423} \frac{(2 + \log(2 \cdot 34423))\log(1/\delta) + 2\log n + 2\log\log\left(1/\Delta_{(s)}^2\right)}{\log(1/\delta) + \log n + \log\log(1/\Delta_{(s)}^2)} \Delta_{(s)}^2$$

$$\leq \frac{640(2 + \log(2 \cdot 34423))}{34423} \Delta_{(s)}^2 \leq \Delta_{(s)}^2/4 \quad ,$$

which proves (19).

Inequality (19) implies

$$|D_{r_1,j}| \geq |\Delta_{(s)}|/2 \geq 4 \cdot \sqrt{\frac{4}{\left\lfloor \frac{2^k}{n} \right\rfloor} \log\left(\frac{nk^3}{0.15\delta}\right)}$$

and by points 1 and 2 we have a guarantee that

$$\left|\hat{D}_{r_1,j}\right| \geq 3 \cdot \sqrt{\frac{4}{\left\lfloor \frac{2^k}{n} \right\rfloor} \log\left(\frac{nk^3}{0.15\delta}\right)} \quad .$$

By line 8 of Algorithm 3, this is sufficient for the algorithm to terminate after the $L^{\text{th}}$ round of iteration $k$.

**Algorithm 3 clusters correctly**  Consider the first $k' \in \mathbb{N}$ with $k' > \log_2(n)$ such that for the samples

$$X_{r_0,j}^{(1)}, \ldots, X_{r_0,j}^{(\lfloor 2^{k'}/n \rfloor)} \sim^{\text{i.i.d.}} \nu_{r_0,j} \quad \text{and} \quad X_{r_1,j}^{(1)}, \ldots, X_{r_1,j}^{(\lfloor 2^{k'}/n \rfloor)} \sim^{\text{i.i.d.}} \nu_{r_1,j}$$

we have that

$$\frac{1}{\lfloor 2^{k'}/n \rfloor}\left|\sum_{t=1}^{\left\lfloor 2^{k'}/n \right\rfloor} X_{r_1,j}^{(t)} - X_{r_0,j}^{(t)}\right| > 3 \cdot \sqrt{\frac{4}{\left\lfloor \frac{2^{k'}}{n} \right\rfloor} \log\left(\frac{nk'^3}{0.15\delta}\right)} \quad .$$

Then by line 8, we know that after completing the iteration Algorithm 3 terminates. From point 2 we know that on $\xi_{\text{cbc}}$ it holds

$$|D_{r_1,j}| > 2 \cdot \sqrt{\frac{4}{\left\lfloor \frac{2^{k'}}{n} \right\rfloor} \log\left(\frac{nk'^3}{0.15\delta}\right)} \ .$$

So if for each $i \geq 2$ we sample again

$$X_{r_0,j}^{(1)}, \ldots, X_{r_0,j}^{(\lfloor 2^{k'}/n \rfloor)} \sim^{\text{i.i.d.}} \nu_{i,j} \quad \text{and} \quad X_{i,j}^{(1)}, \ldots, X_{i,j}^{(\lfloor 2^{k'}/n \rfloor)} \sim^{\text{i.i.d.}} \nu_{i,j} \ ,$$

then for the averages holds again by point 2 that

$$\frac{1}{\lfloor 2^{k'}/n \rfloor} \left| \sum_{t=1}^{\lfloor 2^{k'}/n \rfloor} X_{i,j}^{(t)} - X_{r_0,j}^{(t)} \right| > \sqrt{\frac{4}{\left\lfloor \frac{2^{k'}}{n} \right\rfloor} \log\left(\frac{nk'^3}{0.15\delta}\right)}$$

if and only if $D_{i,j} \neq 0$. So on $\xi_{\text{cbc}}$, the labeling in line 12 yields to a perfect clustering $\hat{g} = g$.

**Bounding the budget:** Similar to the proof of Theorem E.1, we can bound

$$\tilde{L}_{\max} \leq 70(\log(d) + \log\log(1/\delta)) \ .$$

and

$$2^L \leq 192\frac{d}{s}(\log\log d + \log(1/\delta)) \ .$$

So again by defining $C := 156 \cdot 70^3 \cdot 192$ and letting $k^*$ being minimal such that

$$2^{k+1} \geq C \min_{s \in [d]} \left( \frac{(\log d + \log\log(1/\delta))^3 (\log\log d + \log(1/\delta))d}{s\Delta_{(s)}^2} \right.$$
$$+ \frac{(\log d + \log\log(1/\delta))(\log\log d + \log(1/\delta))d}{s}$$
$$\left. + \frac{\left(\log(1/\delta) + \log_+ \log\left(1/\Delta_{(s)}^2\right) + \log n\right) \cdot n}{\Delta_{(s)}^2} \right)$$

we know from (17) and (18) that with probability at least $1 - \delta$, Algorithm 3 terminates and clusters correctly, spending a budget of at most

$$\sum_{k=1}^{k^*} \sum_{1 \leq L \leq \tilde{L}_{\max}: \ L \cdot 2^L \leq 2^{k+1}} 2 \cdot 2^{k+1} \leq 8\tilde{L}_{\max}2^{k^*}$$

$$\leq C'(\log d + \log\log(1/\delta)) \min_{s \in [d]} \left[ \left( \frac{(\log d + \log\log(1/\delta))^3(\log\log d + \log(1/\delta))}{\Delta_{(s)}^2} + 1 \right) \frac{d}{s} \right.$$
$$\left. + \left( \frac{\log(1/\delta) + \log_+ \log\left(1/\Delta_{(s)}^2\right) + \log n}{\Delta_{(s)}^2} + 1 \right) n \right] \ ,$$

where $C > 0$ is a numerical constant. Inserting $\tilde{s}$ in the right hand side and gathering the logarithmic terms like in the proof of Proposition E.1 yields the claim. $\qquad\square$

# G. Proof of Theorem C.1 on the extension to $K > 2$

**Come-back on sub-routines CR and CBC.** We can reformulate Proposition E.1 in the following corollary.

**Corollary G.1.** *Let $\delta \in (0, 1/e)$, $r \in [n]$ and $G \subset [n]$. There exist an event of probability larger than $1 - \delta$ such that*

1. *If $M_{i,\cdot} = M_{r,\cdot} \; \forall i \in G$, then $CR(\delta, r, G)$ does not stop.*

2. *If $\exists i \in G$ such that $M_{i,\cdot} \neq M_{r,\cdot}$, then $CR(\delta, r, G)$ returns an item $s$ such that $M_{r,\cdot} \neq M_{s,\cdot}$, with a budget $T$ smaller than*
$$\tilde{C} \log\left(\frac{1}{\delta}\right) \min_{h > 0} \left[\frac{d|G|}{|\{(i,j) \in G \times [d] \; ; \; |M_{r,j} - M_{i,j}| \geqslant h\}|} \left(\frac{1}{h^2} + 1\right)\right] \; ,$$
   *with $\tilde{C}$ a poly-logarithmic term defined in Proposition E.1.*

Proposition F.1 can be formulated as follows:

**Corollary G.2.** *Let $\delta \in (0, 1/e)$, $r \in [n]$, $s \in [n]$ and $G \subset [n]$. Assume that $M_{r,\cdot} \neq M_{s,\cdot}$, then there exists an event of probability larger than $1 - \delta$ such that $CBC(\delta, r, s; G)$ outputs a partition of $G$, $G = R \sqcup S$ with a budget $T$ such that*

1. $\{i \in G \; ; \; M_{i,\cdot} = M_{r,\cdot}\} \subset R$ *and* $\{i \in G \; ; \; M_{i,\cdot} = M_{s,\cdot}\} \subset S$

2. *the budget $T$ is smaller than*
$$\tilde{C} \log\left(\frac{1}{\delta}\right) \min_{h} \left[\left(\frac{d}{|\{j \in [d] \; ; \; |M_{r,j} - M_{s,j}| \geqslant h\}|} + n\right) \left(\frac{1}{h^2} + 1\right)\right] \; ,$$
   *with $\tilde{C}$ a poly-logarithmic term defined in Proposition F.1.*

**Proof of Theorem C.1.** We use the notation from the pseudo-code Algorithm 5 and from Appendix C. The correction of the algorithm is a direct consequence of the following lemma, which states that, with high probability, all epochs behave as expected. The bound on the total budget given in Theorem C.1 follows directly by summing the sample complexities of all CR and CBC calls across the $K - 1$ epochs. These individual complexities are provided in Corollaries G.1 and G.2.

**Lemma G.3.** *There exists an event of probability larger than $1 - \delta$ such that for each epoch $1 \leqslant e \leqslant K - 1$,*

1. *Epoch $e$ terminates using a finite budget.*

2. *The item $r_{e+1}$ selected is a new representative: $M_{r_{e+1},\cdot} \notin \{M_{r_1,\cdot}, \ldots, M_{r_e,\cdot}\}$*

3. *For all $k \in [e + 1]$, $\{i \in [n]; \; M_{i,\cdot} = M_{r_k,\cdot}\} \subset \hat{G}_k^{(e+1)}$*

Let $\mathcal{E}$ denote the event that, for all epochs $e = 1, \ldots, K - 1$, each call to CR and CBC behaves correctly — i.e., satisfies Points 1 and 2 of Corollaries G.1 and G.2. In epoch $e$, the algorithm makes $e$ calls to CR and $e$ calls to CBC, each with a confidence level $\delta_e = \frac{\delta}{e(K-1)}$. By a union bound over all calls across all epochs, the probability of event $\mathcal{E}$ is at least $1 - \delta$.

We prove by induction on $e \in [K - 1]$ that the three points of Lemma G.3 hold on $\mathcal{E}$.

Initially, we have a trivial partition $G^{(1)} = [n]$, and a representative $r_1$ is arbitrarily selected. Points 1–3 trivially hold at $e = 1$.

Assume that, at epoch $e$, $(r_1, \ldots, r_e)$ are representatives from $e$ distinct clusters, and for each $k \in [e]$, the set $\hat{G}_k^{(e)}$ contains all items with mean $M_{r_k,\cdot}$.

Since $e < K$, there exists at least one cluster not yet represented, and therefore, there exists some $i \in \hat{G}_k^{(e)}$ and some $k \in [e]$ such that $M_{i,\cdot} \neq M_{r_k,\cdot}$. On event $\mathcal{E}$, $CR(\delta_e, r_k; G_k^{(e)})$ will return an item in finite time, and Line 5 of the corresponding algorithm terminates. Denote $r_{e+1}$ the item returned from this call. Then, by Corollary G.1, we know that $M_{r_{e+1},\cdot} \neq M_{r_k,\cdot}$.

Moreover, for $k' \neq k$, $G_{k'}^{(e)}$ contains all items with mean $M_{r'_k,\cdot}$ so that $M_{r_{e+1},\cdot} \notin \{M_{r_1,\cdot}, \ldots, M_{r_e,\cdot}\}$, and Point 2 also holds for the $e$-th epoch. Since all calls terminate in finite time on $\mathcal{E}$, Point 1 also holds.

Now, for each $k \in [e]$, $M_{r_{e+1},\cdot} \neq M_{r_k,\cdot}$, so that on $\mathcal{E}$, the partition of $\hat{G}_k^e$ into two groups $\hat{G}_k^{e+1} \sqcup \hat{R}_k^{e+1}$ will perfectly separate the items with mean $M_{r_k,\cdot}$ and $M_{r_{e+1},\cdot}$. By assumption, the partial cluster $\hat{G}_k^e$ already contains all items with mean $M_{r_k,\cdot}$, so $\{i \in [n] ; M_{i,\cdot} = M_{r_k,\cdot}\} \subset \hat{G}_k^e$. Similarly, as $\hat{G}_1^e, \ldots, G_e^{(e)}$ is a partition of $[n]$, any item $i$ with mean $M_{r_{e+1},\cdot}$ will be set in one of the sets $\hat{R}_k^{e+1}$ so that $\{i \in [n] ; M_{i,\cdot} = M_{r_{e+1},\cdot}\} \subset \cup_{k=1}^e \hat{R}_k^{e+1} = \hat{G}_{e+1}^{e+1}$. Thus, Point 3 also holds for epoch $e$.

By induction, all three points in the lemma hold for all $e \in [K-1]$ on event $\mathcal{E}$, which concludes the proof. $\qquad \square$

## H. Proof of the lower bounds

The lower bound in Theorem 4.1 consists of two terms, which we prove separately. In the proofs, we use $T_{i,j}$ as the number of time a procedure selects the pair $(i, j) \in [n] \times [d]$.

**Lemma H.1.** *The $(1 - \delta)$-quantile of the budget of any $\delta$-PAC algorithm $\mathcal{A}$ is bounded as follows*

$$\max_{\tilde{\nu} \in \mathcal{E}_{per}(M)} \mathbb{P}_{\tilde{\nu},\mathcal{A}} \left( T \geqslant \frac{2d}{\theta \|\Delta\|_2^2} \log \frac{1}{6\delta} \right) \geqslant \delta . \tag{20}$$

*Proof of Lemma H.1.* Fix an algorithm $\mathcal{A}$, and let $\mathcal{E}_{per}(M)$ denote the set of Gaussian environments obtained by permuting the rows and columns of $M$. For the purpose of the proof, we define $\mathbb{P}\sigma, \tau$ as the probability distribution induced by the interaction between algorithm $\mathcal{A}$ and the environment defined in (4), where $\sigma$ and $\tau$ are permutations of the rows and columns of $M$, respectively.

We permute the rows of $M$ to reflect the fact that the learner has no access to the label vector $g$. In addition, we permute the columns of $M$ to account for the algorithm's ignorance of the structure of the gap vector $\Delta$, in particular, the identity of the feature with the largest gap.

Without loss of generality, we assume that $\mu_0 = \mathbf{0}$ and $\mu_1 = \Delta$, with the group associated with mean vector $\Delta$ being the smaller of the two.

Define $\chi$ as the smallest integer such that for all permutations $\sigma$ and $\tau$ of $1, \ldots, n$ and $1, \ldots, d$, respectively, the following inequality holds:

$$\mathbb{P}_{\sigma,\tau}(T > \chi) \leqslant \delta . \tag{21}$$

Our goal is to derive a lower bound on $\chi$. Intuitively, we show that for small $\chi$, there exists a permutation of $M$ for which it is impossible to detect a nonzero entry.

Introduce $\mathbb{P}_0$ as the probability distribution induced by $\mathcal{A}$, in an environment where all items belong to a single cluster, i.e., each $X_t \sim \mathcal{N}(0, 1)$. We will prove that under this "null" environment, the algorithm $\mathcal{A}$ requires more than $\chi$ samples with probability at least $1 - 2\delta$.

To this end, consider an environment $\nu(g, \mu)$ consisting of two clusters with means $\mathbf{0}$ and $\mu$, and let $\mu \to 0$. Since $\mathcal{A}$ is $\delta$-PAC, there exist two distinct partitions $g \neq g'$ and an event $A$ such that

$$\mathbb{P}_{\nu(g,\mu)}(A, T \leqslant \chi) + \mathbb{P}_{\nu(g',\mu)}(A^c, T \leqslant \chi) \leqslant 2\delta .$$

For example, take $g(1) = 0$, $g(2) = 1$, and $g'(1) = 0$, $g'(2) = 0$, the event $\{\hat{g}(1) = \hat{g}(2)\}$ suffices. Then, conditionally on $T \leqslant \chi$, $\mathbb{P}_{\nu(g,\mu)}$ and $\mathbb{P}_{\nu(g',\mu)}$ converge in total variation to $\mathbb{P}_0$ as $\mu \to 0$. consider an environment $\nu(g, \mu)$ consisting of two clusters with means $\mathbf{0}$ and $\mu$, and let $\mu \to 0$.

$$\mathbb{P}_0(T \leqslant \chi) \leqslant 2\delta . \tag{22}$$

Applying the Bretagnolle–Huber inequality (see Lattimore & Szepesvári, 2020, Thm. 14.2), and combining (21) and (22), we obtain

$$\frac{1}{2} \exp\left(-\operatorname{KL}(\mathbb{P}_0, \mathbb{P}_{\sigma,\tau})\right) \leqslant \mathbb{P}_0(T \leqslant \chi) + \mathbb{P}_{\sigma,\tau}(T > \chi) \leqslant 3\delta ,$$

which implies

$$\log \frac{1}{6\delta} \leqslant \operatorname{KL}(\mathbb{P}_0, \mathbb{P}_{\sigma,\tau}) . \tag{23}$$

Next, using the decomposition of KL divergence for bandit models (see Lattimore & Szepesvári, 2020, Lemma. 15.1), and the Gaussian assumption, we have

$$\text{KL}(\mathbb{P}_0, \mathbb{P}_{\sigma,\tau}) = \sum_{i,j} \mathbb{E}_0[T_{i,j}] \, \text{KL}(\mathbb{P}_0^{i,j}, \mathbb{P}_{\sigma,\tau}^{i,j}) = \sum_{i,j} \mathbb{E}_0[T_{i,j}] \mathbb{1}_{g(\sigma(i))=1} \frac{\Delta_{\tau(j)}^2}{2} \quad . \tag{24}$$

Averaging both sides of (23) over all permutations $\sigma, \tau$, and using Equations (23) and (24), we get

$$\log \frac{1}{6\delta} \leqslant \frac{1}{n!} \frac{1}{d!} \sum_{\sigma,\tau} \mathbb{E}_0[T_{i,j}] \mathbb{1}_{g(\sigma(i))=1} \frac{\Delta_{\tau(j)}^2}{2} \quad . \tag{25}$$

Now, observe that each element in $i \in \{1, \ldots, n\}$ (resp. $j \in \{1, \ldots, d\}$) appears exactly $(n-1)!$ (resp. $(d-1)!$) times in the multi-set $\{\sigma(i)\}_\sigma$ (resp. $\{\tau(j)\}_\tau$), so that

$$\frac{1}{n!} \frac{1}{d!} \sum_{\sigma,\tau} \sum_{i,j} \mathbb{E}_0[T_{i,j}] \mathbb{1}_{g(\sigma(i))=1} \frac{\Delta_{\tau(j)}^2}{2} = \frac{(n-1)!}{n!} \frac{(d-1)!}{d!} \sum_{k,l} \sum_{i,j} \mathbb{E}_0[T_{i,j}] \mathbb{1}_{g(k)=1} \frac{\Delta_l^2}{2}$$

$$= \frac{1}{n} \sum_{k \in [n]} \mathbb{1}_{g(k)=1} \frac{\|\Delta\|_2^2}{2d} \mathbb{E}_0[T] \quad .$$

Since the group associated with $\Delta$ is the smallest, $\frac{1}{n} \sum_{k \in [n]} \mathbb{1}_{g(k)=1} = \theta$. Using a modified algorithm $\mathcal{A}'$ that stops at $T \wedge \chi$, we can bound $\mathbb{E}_0[T] \leqslant \chi$. Finally, it follows that:

$$\chi \geqslant \frac{2d}{\theta \|\Delta\|_2^2} \log \frac{1}{6\delta} \quad .$$

Since $\chi$ is the maximum over all permuted environments constructed with $M$ of the $(1-\delta)$-quantile of the budget, this inequality concludes the proof of Lemma H.1. $\qquad\square$

**Lemma H.2.** *Assume that $\delta < 1/2$. If $\mathcal{A}$ is $\delta$-PAC for the clustering problem, then for any environment $\nu$,*

$$\mathbb{E}_{\mathcal{A},\nu}[T] \geqslant \frac{2(n-2)}{\Delta_{(1)}^2} \log \left( \frac{1}{2.4\delta} \right) \quad , \tag{26}$$

*where $|\Delta_{(1)}| = \max_{j \in [d]} |\Delta_j|$.*

*Proof of Theorem 4.1.* Observe that Lemma H.2 does not directly provide a high-probability lower bound on the budget. We now show how the expectation bound given by Lemma H.2 implies a lower bound on the $(1-\delta)$-quantile of the budget.

Let $\mathcal{A}$ be any $\delta$-PAC algorithm. Assume, by contradiction, that

$$\mathbb{P}_{\nu,\mathcal{A}} \left( T \geqslant \frac{2(n-2)}{\Delta_{(1)}^2} \log \left( \frac{1}{4.8\delta} \right) \right) < \delta.$$

We modify $\mathcal{A}$ such that it stops at time

$$T' := T \wedge \frac{2(n-2)}{\Delta_{(1)}^2} \log \left( \frac{1}{4.8\delta} \right) \quad .$$

If $\mathcal{A}$ reaches time $\frac{2(n-2)}{\Delta_{(1)}^2} \log \left( \frac{1}{4.8\delta} \right)$, it stops sampling and outputs an error. The resulting algorithm $\mathcal{A}'$ is $2\delta$-PAC, with a budget satisfying

$$\mathbb{E}_{\mathcal{A}',\nu}[T'] \leqslant \frac{2(n-2)}{\Delta_{(1)}^2} \log \left( \frac{1}{2.4\delta} \right) \quad .$$

However, this contradicts Lemma H.2, applied to $\mathcal{A}'$ with $\delta' = 2\delta$. Thus, we have $\mathbb{P}_{\nu,\mathcal{A}} \left( T \geqslant \frac{2(n-2)}{\Delta_{(1)}^2} \log \left( \frac{1}{4.8\delta} \right) \right) \geqslant \delta$ . $\quad\square$

*Proof of Lemma H.2.* Let $\mathcal{A}$ be any $\delta$-PAC algorithm for the clustering problem, and consider the matrix $M$ that parametrizes the Gaussian environment $\nu$. We fix for all environments in this proof that $g(1) = 0$ and $g(2) = 1$. It implies intuitively that we assume that the algorithm knows one item from each group via an oracle.

For the Gaussian environment $\nu$, let $i, j \in [n] \times [d]$. The observations follow a Gaussian distribution:

$$\nu_{i,j} = \begin{cases} \mathcal{N}(0, 1), & \text{if } g(i) = 0, \\ \mathcal{N}(\Delta_j, 1), & \text{if } g(i) = 1 \ . \end{cases}$$

We aim to show that with a budget smaller than $\frac{cn}{\Delta_{(1)}^2} \log(1/\delta)$, a $\delta$-PAC algorithm cannot distinguish the environment $\nu$ from another environment where one item from $\nu$ has been switched to the other group. We construct now this alternative environment.

For any $k \in \{3, \ldots, n\}$, define $g^k$ as the vector of labels obtained from $g$ by flipping the label of row $k$, and let $\nu^k$ denote the corresponding Gaussian environment. The lower bound follows from the information-theoretic cost of distinguishing $\nu$ from any $\nu^k$.

To handle multiple environments, let $\mathbb{P}_{g^k}$ (resp. $\mathbb{P}_g$) denote the probability distribution induced by the interaction between algorithm $\mathcal{A}$ and environment $\nu^k$ (resp. $\nu$).

For any $k \in \{3, \ldots, n\}$, note that environments $\nu$ and $\nu^k$ differ only on row $k$. By decomposing the KL divergence and using the Gaussian KL formula, we have:

$$\mathrm{KL}(\mathbb{P}_g, \mathbb{P}_{g^k}) = \sum_{j=1}^{d} \mathbb{E}_g[T_{k,j}] \frac{\Delta_j^2}{2} \leqslant \sum_{j=1}^{d} \mathbb{E}_g[T_{k,j}] \frac{\Delta_{(1)}^2}{2} \ , \tag{27}$$

where we use that $|\Delta_{(1)}| = \max_{j \in [d]} |\Delta_j|$, and $T_{k,j}$ denotes the number of samples taken from row $k$ and column $j$.

Since $\mathcal{A}$ is $\delta$-PAC for the clustering task, we have:

$$\mathbb{P}_g(\hat{g} \neq g) \leqslant \delta, \qquad\qquad \mathbb{P}_{g^k}(\hat{g} \neq g^k) \leqslant \delta \ .$$

Now, if $\delta \in (0, 1/2)$, by the monotonicity of the binary KL divergence $\mathrm{kl}$, and using the data-processing inequality, we obtain:

$$\mathrm{kl}(\delta, 1 - \delta) \leqslant \mathrm{kl}\left(\mathbb{P}_g(\hat{g} = g^k), \mathbb{P}_{g^k}(\hat{g} = g^k)\right) \leqslant \mathrm{KL}(\mathbb{P}_g, \mathbb{P}_{g^k}) \ . \tag{28}$$

Combining Equation (27) and Equation (28), and summing over $k \in \{3, \ldots, n\}$, we get:

$$(n-2) \, \mathrm{kl}(\delta, 1 - \delta) \leqslant \sum_{k=3}^{n} \sum_{j=1}^{d} \mathbb{E}_g[T_{k,j}] \frac{\Delta_{(1)}^2}{2} \leqslant \mathbb{E}_g[T] \frac{\Delta_{(1)}^2}{2} \ . \tag{29}$$

Finally, Lemma H.2 follows by combining Equation (29) with the inequality $\mathrm{kl}(\delta, 1 - \delta) \geqslant \log\left(\frac{1}{2.4\delta}\right)$. $\qquad\square$

# I. Technical Results

**Lemma I.1** (Chernoff-Bound for Binomial random variables). *For $i = 1, \ldots, n$, consider $X_1, X_2, \ldots, X_n \sim^{\text{i.i.d.}} \mathrm{Bern}(p)$ with $p \in (0, 1)$, denote $\mu := np$ and consider $\kappa > 0$. We have*

$$\mathbb{P}\left(\sum_{i=1}^{n} X_i \geq (1 + \kappa)\mu\right) \leq \frac{e^{\kappa\mu}}{(1 + \kappa)^{(1+\kappa)\mu}}.$$

*If $\kappa \in (0, 1)$, we also have*

$$\mathbb{P}\left(\sum_{i=1}^{n} X_i \leq (1 - \kappa)\mu\right) \leq \exp\left(-\frac{\kappa^2 \mu}{2}\right).$$

