# OpenReview forum: "Clustering Items through Bandit Feedback: Finding the Right Feature out of Many"
_ICML.cc/2025/Conference — ICML 2025 poster_

### Official Review · Reviewer_D6WK · 2025-03-09

**Overall Recommendation:** 3

**Summary:**

This paper studies a problem of clustering $n$ items into two groups using bandit feedback. This setting considers an $n \times d$ matrix $M$, where each row represents an item's feature vector. The $n$ rows are partitioned into two unknown groups, such that items within the same group share the same feature vector. The learner sequentially and adaptively an item $I_t \in [n]$ and a feature $J_t \in [d]$, and gets a noisy observation drawn from an unknown distribution with mean $M_{I_t,J_t}$. The goal is to recover the correct item partition with minimal observations. The authors propose an algorithm called `BanditClustering`, which operates in two steps: (1) identifying two representative items from different groups, and (2) selecting a discriminative feature to classify all items. The authors provide theoretical results such as a tight upper bound on the required budget and a matching instance-dependent lower bound. Numerical experiments validate the efficiency of the method compared to non-adaptive clustering approaches.

## update after rebuttal
Overall, I appreciate the technical novelty of this paper. Therefore, I decided to maintain my positive score.

**Claims And Evidence:**

All claims are well-supported by clear and convincing evidence.

**Essential References Not Discussed:**

N/A.

**Experimental Designs Or Analyses:**

The experiments rely solely on synthetic data and consider only uniform sampling and K-means as baselines. It would be better to compare other adaptive clustering methods, such as the ones mentioned in the related work. Also, the comparison does not examine performance in highly imbalanced scenarios (when $\theta$ is small).

**Methods And Evaluation Criteria:**

The proposed methods and evaluation criteria make sense for the problem.

**Other Comments Or Suggestions:**

Quotation marks in LaTeX should be formatted using `` and '' instead of " ".

The legend in Figure 1 is unclear. What does the algorithm "Cluster" refer to? Also, it seems that the uniform sampling algorithm is not explicitly labeled. Furthermore, what does the shaded area indicate?

**Other Strengths And Weaknesses:**

Strengths
1. This paper is well-organized and easy to follow.
2. This paper studies a new bandit clustering problem where at each step, only a single noisy feature of a given item is observed. This setting is different from previous studies where the entire noisy feature vector is observed.
3. The authors provide rigorous upper and lower bounds on sample complexity, ensuring that their method is provably efficient.

Weaknesses
1. The algorithms and theoretical analysis assume only two clusters, which is a strong limitation. While the authors suggest that they could "straightforwardly extend" their methodology to multiple clusters, this extension is not formally developed or analyzed.
2. The paper primarily focuses on theoretical contributions and lacks strong motivation. Although the introduction includes a motivating example, it does not clearly articulate the significance of the problem.

**Questions For Authors:**

1. Why do the authors restrict their analysis to only two clusters? What challenges arise when extending the approach to multiple clusters?
2. How does the proposed algorithm perform when the data is imbalanced (i.e., $\theta$ is small), and how does it compare to the baselines?

**Relation To Broader Scientific Literature:**

This work extends the bandit clustering problem studied in prior works, such as Thuot et al. (2024) and Yang et al. (2024). Unlike these studies, where the entire noisy feature vector of an item is observed at each step, the setting introduced in this paper allows only a single noisy feature to be observed per step. This introduces a new and inherently more challenging problem.

**Theoretical Claims:**

I have not checked all the proofs in detail but did not identify any obvious errors.

---

> ### Author Rebuttal · Authors · 2025-03-31
>
> First, we would like to thank you for your time and effort in reviewing our paper. Below, we address the key remarks and questions raised in your review:
>
> - **Extension to $K>2$ clusters**
>
>   In the paper, we analyze the case of two clusters, as even in this simpler setting, significant challenges appeared to obtain optimality. It happens that it is possible to use our algorithms as subroutines in order to handle the extension to $K$ cluster. To avoid redundancy, we invite you to read the answer to reviewer RdCv where this extension is discussed in detail. When dealing with $K>2$ groups, it is significantly more challenging to find a strategy which adapts optimally to the relative position of the centers of the $K$ groups, we will leave this question for future works.
> - **Dependency on the balancedness $\theta$**
>
>   Thank you for the question.  From a theoretical point,  we do prove in the paper that the balancedness parameter $\theta$ has only an impact on the first step of our procedure, i.e., the identification of representatives.  Furthermore, gathering Proposition C.1 (analysis of Algorithm 2), and the lower bound in Lemma E.1, we are able to prove that our procedure exhibits the optimal dependency on the budget with respect to $\theta$.
>   The only difficulty to deal with very unbalanced clusters is that it is hard to detect a row in the smallest cluster. We will add further discussion about the dependency on $\theta$ of our method in the final paper.
>
>   We have also run numerical experiments to investigate the influence of $\theta$. They will be included in the paper. In a first setup, we are able to see that indeed, $\theta$ mainly influences the CandidateRow-step of our clustering procedure and that the influence on the budget of this step is comparable to a factor $1/\theta$, which fits in with our theoretical findings. Furthermore we can see that in this setup, our algorithm clearly outperforms uniform allocation strategies for very small values of $\theta$, while being competitive for larger values of $\theta$.
>
> - **Suggestions**
>
>   Thank you to your suggestion, we will correct all quotation marks, and improve the explanation and legend of Figure 1.
> - **Motivation**
>
>   We thank the reviewer for pointing out that we should extend the discussion of our motivating example, which comes from image labeling. Recent work about distinguishing "doppelganger" animals  uses expert annotators to ensure a correct labeling (see [Herde, Marek, et al. "dopanim: A Dataset of Doppelganger Animals with Noisy Annotations from Multiple Humans." Advances in Neural Information Processing Systems 37 (2024): 51085-51117.]). This works because experts know which features are relevant for a correct distinction. If no experts are available, our approach still provides a solution, because we can find the relevant features without prior knowledge and use non-expert workers for the classification task.
>
> - **Numerical experiments**
>
>   As mentioned above, we will illustrate the dependency with $\theta$ in an additional numerical experiment. Besides, we will also compare our method with the method from [Ariu et al.(2024) "Optimal clustering from noisy binary feedback"]).

---

### Official Review · Reviewer_RdVc · 2025-03-12

**Overall Recommendation:** 4

**Summary:**

This paper investigates the problem of clustering items via bandit feedback. The items can be partitioned into two unknown groups that share the same feature vector within each cluster. The authors propose a sequential and adaptive setting where the learner can only select one item-feature pair per round. The objective is to accurately recover the correct partition while minimizing the number of observations. The paper presents an algorithm that identifies a relevant feature for clustering by leveraging the Sequential Halving algorithm. With probability at least $1-\delta$, the algorithm achieves accurate recovery of the partition, and the authors derive an upper bound on the required budget. Additionally, they establish an instance-dependent lower bound that is tight in certain relevant cases.

**Claims And Evidence:**

Yes

**Essential References Not Discussed:**

There is another line of clustering of bandits’ works. For example:

1. Gentile, Claudio, Shuai Li, and Giovanni Zappella. "Online clustering of bandits." In International conference on machine learning, pp. 757-765. PMLR, 2014.
2. Li, Shuai, Wei Chen, and Kwong-Sak Leung. "Improved algorithm on online clustering of bandits." arXiv preprint arXiv:1902.09162 (2019).
3. Liu, Xutong, Haoru Zhao, Tong Yu, Shuai Li, and John CS Lui. "Federated online clustering of bandits." In Uncertainty in Artificial Intelligence, pp. 1221-1231. PMLR, 2022.
4. Li, Zhuohua, Maoli Liu, Xiangxiang Dai, and John Lui. "Demystifying Online Clustering of Bandits: Enhanced Exploration Under Stochastic and Smoothed Adversarial Contexts." arXiv preprint arXiv:2501.00891 (2025).

**Experimental Designs Or Analyses:**

Yes.

**Methods And Evaluation Criteria:**

Yes

**Other Comments Or Suggestions:**

## Minor Issues:

There are several typographical errors throughout the paper. For example, on line 230, a sentence is incomplete.

**Other Strengths And Weaknesses:**

## Weaknesses:

1. In Section 6, the authors discuss two important extensions: increasing the number of groups ($K>2$) and handling heterogeneous groups, which would better model real-world applications. However, the discussion provides only vague ideas without specific results for each extension. More concrete analysis or preliminary results would strengthen the paper's practical relevance.
2. Regarding the upper bound result, there appears to be a factor of $d$ in the regret upper bound in Equation (3). It seems possible that fixing one feature to explore could remove this d factor, potentially at the cost of a larger $\Delta$ term. The paper would benefit from a more thorough discussion of this trade-off, especially in cases where the d improvement might dominate any negative effects on the $\Delta$ term.

**Questions For Authors:**

Please comment/discuss the weakness part. I am happy to raise my score if these questions are well discussed/solved.

-------------Post-rebuttal----------------

The reviewer thanks the authors for their detailed responses, including the discussion on extending to more than two groups ($K > 2$), addressing heterogeneous groups, elaborating on the trade-offs between $d$ and $\Delta$, and discussing the missing references. My concerns have been resolved, and I would like to raise my score. I encourage the authors to incorporate these discussions into the final version to improve clarity and completeness of the current work.

**Relation To Broader Scientific Literature:**

## Contributions:

1. Introduction of a new bandit cluster model where only one item-feature pair can be selected in each round, distinguishing it from existing works that select an item with all features.
2. Development of the first lower bound that characterizes the fundamental difficulty of the bandit clustering problem.
3. Design of a near-optimal bandit clustering algorithm with theoretical guarantees.

**Theoretical Claims:**

Yes, I checked the main content of the paper.

---

> ### Author Rebuttal · Authors · 2025-03-31
>
> First, we thank you for your time and effort in reviewing our paper.  We will correct the typographical errors you identified. Below, we discuss about the different weaknesses that you identified. We will provide intuitions and thorough discussions on these points in the final version of the paper.
> - **Extension to $K>2$ clusters**
>
>   Our algorithm identifies a discriminative feature to separate two groups.  To achieve this. We fix arbitrarily a first item, and find an item from a different cluster (Algorithm 2). Then, we identify a feature that separates the two clusters well, balancing the cost of identifying such a feature with the cost of classification (Algorithm 3). We proved that this requires a budget of $H(\mu^a-\mu^b)\log(1/\delta)$ (see Thm 3.1 and Eq. (3)).
>
>   Here is a way to extend the method to $K>2$ clusters, assuming $K$ is known. We first identify $K$ representatives from different clusters, one at a time. If $k<K$ items have already been identified from $k$ different clusters, we can run $k$ calls of Algorithm 1 in parallel to identify an item whose mean vector differs from all the previously identified representatives. Repeating this operation $K$ times yields $K$ representatives. Then, we use these representatives to learn $K(K-1)/2$ discriminative features (one for each pair of representatives), ultimately classifying all items. Compared to the case where $K=2$, the total budget then scales by a factor of $K^2 \times \min_{k\ne k'} H(\mu^k-\mu^{k'})$, where $\mu^1,\dots,\mu^K$ are the mean vectors of the $K$ clusters. This approach would be order-optimal if $K$ is considered a constant.
>
> We will add this procedure for $K>2$ clusters in the Appendix of the final version, along with this (non-optimal) bound.  However, finding a distribution-dependent optimal strategy, taking into account the relative positions of all means $\mu^1,\dots, \mu^K$, remains highly non-trivial and is way beyond the scope of our paper.
> - **Robustness to heterogeneity within the clusters**
>
>   Actually, our method can be adapted to deal with problems where there exists some heterogeneity inside the groups. We invite you to read the answer in the rebuttal to reviewer AdRS.
> - **Intuition on the complexity $H$ for sparse vector**
>
>   We would like to provide more intuition on the trade-off $H$ (see Eq. (3)). In particular, we will explain why a dependency in the dimension $d$ is unavoidable and that $H$ is the optimal trade-off between $d$ and $\Delta$.
>
>   Most of our intuitions about the problem lies in the simple case (Corollary 3.2), where the gap vector is $s$-sparse with a constant magnitude $h$. In this case, the l2 norm of the gap vector $\Delta$ (appearing in $H$) becomes $||\Delta||_2^2=sh^2$. Before explaining the intuition behind this rate, we emphasize that the lower bound matches Theorem 4.1 in this setting thereby showing that our budget is indeed optimal.
>
>   To perform the clustering task, we need to select a feature that discriminates the two groups. if one item in each cluster are identified , this problem is equivalent to finding a good arm of reward $h$ among $d$. In this case, a simple algorithm --which is at the core of our approach-- proceeds as follows.
>    - First, sub-sample $\frac{d}{s\theta}\log(1/\delta)$ entries from the matrix, so that, with high probability, a nonzero entry is selected.  If $s=d$, the problem reduces to one dimension; if $s=1$, all features must be explored.
>   - Second, sample all these entries $\frac{1}{h^2}\log(1/\delta)$ times, select the entry with the largest (absolute) mean, and store the associated feature.
>   - Finally, sample  $\frac{1}{h^2}\log(1/\delta)$ each item on this feature and classify the items.
>
>   In the paper, our algorithm (especially through the use of sequential halving) adapts to the unknown parameters $s$ and $h$, and leads to a budget $\frac{d}{\theta sh^2}\log(1/\delta)^2+\frac{n}{h^2}\log(1/\delta)$. We can argue that all these steps cannot be improved with respect to $d,s,h$. We will provide more intuition on this optimal trade-off in the final version. Also, you can find a discussion on the optimality of $H$ for general shapes of $\Delta$ in the rebuttal of reviewer z8W5.
> - **Literature on online clustering of bandits**
>
>   We appreciate your suggestion regarding references on the problem of online clustering of bandits and its extensions. This problem is indeed related to our setting, as it involves exploring a bandit environment where the items exhibit an underlying clustering structure. We will include a discussion on the similarities and differences between our approach and this interesting line of research in the literature review of our paper. Still, we would like to point out there are two major differences with our problem: (1) the learner has no control over which items are presented at each time step, and (2) the algorithms (such as CLUB and its extensions) are evaluated in the cumulative regret setting.

---

### Official Review · Reviewer_AdRS · 2025-03-15

**Overall Recommendation:** 4

**Summary:**

This paper addresses the problem of clustering items based on bandit feedback in a sequential and adaptive setting. Each of $n$ items is characterized by a $d$-dimensional feature vector, and the items are partitioned into two unknown groups where items within the same group share the same feature vector. The learner sequentially selects an item and a feature, observes a noisy evaluation, and aims to recover the correct partition with a small number of observations.

The main contribution is the BanditClustering procedure, which operates in three steps:

- Identifying two items from distinct groups
- Finding a discriminative feature between them
- Using this feature to cluster all items.

The authors provide non-asymptotic bounds on the budget required and prove their algorithm's optimality through matching lower bounds in certain cases. The approach leverages techniques from best arm identification (Sequential Halving) and adaptive sensing strategies.

The paper presents a theoretically sound algorithm for clustering with bandit feedback that achieves optimal sample complexity by efficiently identifying discriminative features.

The paper includes theoretical analysis characterizing the sample complexity based on the difficulty of the clustering task, which is quantified by the difference between feature vectors of the two groups. Experimental results demonstrate the advantage of their adaptive approach over uniform sampling, particularly in sparse regimes.

**Claims And Evidence:**

The main claims of the paper are supported by convincing theoretical and empirical evidence:

- The upper bound on sample complexity (Theorem 3.1) is rigorously proven and demonstrates how the algorithm adapts to the sparsity pattern of the gap vector.

- The lower bounds (Theorem 4.1) establish the optimality of the approach in certain regimes, particularly when the gap vector takes only two values.

- The experimental results in Section 5 supports the theoretical findings, showing the algorithm's superior performance compared to uniform sampling in sparse regimes.

The claims about the algorithm's adaptivity to unknown structures are well-supported by the theoretical analysis, which shows how the budget scales with problem-dependent parameters without requiring prior knowledge of these parameters.

**Essential References Not Discussed:**

The paper does a good job at comparing with other bandit clustering frameworks but could potentially elaborate more on the quantitative differences in sample complexity. I don't identify any critical omissions in the literature review.

**Experimental Designs Or Analyses:**

Summary of Experimental setup:

- Experiment 1 examines how sparsity affects budget requirements, comparing BanditClustering against uniform sampling with K-means.

- Experiment 2 studies how the budget scales with problem dimension when the sparsity is fixed.

These experiments showcase support to the theoretical claims about adaptivity to sparsity and scaling with problem dimensions. The comparison with uniform sampling is informative, demonstrating clear advantages of the adaptive approach.

However, the experimental section is relatively brief compared to the theoretical analysis. As a proof-of-concept it works, though more detailed experiments wouldn't be a bad addition.

**Methods And Evaluation Criteria:**

The methods and evaluation criteria are generally sound, though there might be room for improving sample efficiency:

- The three stage approach with primary focus on identifying a single feature goes well with the narration of sparse feature distinction vectors.

- The budget (number of observations) is an appropriate metric for evaluating efficiency in the bandit setting.

- The theoretical analysis properly characterizes the algorithm's performance in terms of relevant problem parameters (gap magnitude, sparsity, balancedness).

However, there appears to be potential inefficiency in the sampling strategy, particularly regarding samples from row 1. Maybe a better book-keeping of what samples are already present?

**Other Comments Or Suggestions:**

- The paper would benefit from a more intuitive explanation of why the specific trade-offs in the algorithm design are optimal, perhaps with a simplified example.

- The paper mentions budget comparison with other approaches in the introduction but doesn't provide detailed quantitative comparisons in the main text. (maybe I missed this.)

**Other Strengths And Weaknesses:**

**Strengths:**
The paper's main strengths lie in its theoretical rigor and adaptive approach, while limitations include the restricted problem setup and questions about practical applicability.

- Clearly written paper with well structured arguments and explanation.
- The paper provides a clear theoretical characterization of the sample complexity, showing how it depends on the sparsity pattern of the gap vector.
- The algorithm is adaptive to unknown structures without requiring prior knowledge of the problem parameters.
- The theoretical analysis is rigorous, with matching upper and lower bounds in certain regimes.
- The three-step approach (finding representatives, identifying discriminative features, clustering) is conceptually clear and intuitive.
- The experimental results convincingly demonstrate the advantage over uniform sampling approaches.

**Weaknesses:**
- The setup is limited to binary partitioning (just two clusters) with identical feature vectors within clusters. The scaling up of the computational complexity and regret due to more clusters is something of importance
- The algorithm seems to oversample from row 1, raising questions about sample efficiency. The paper doesn't explore potential downstream effects or implications for regret bounds.
- The practical impact of the theoretical findings on real-world applications is not extensively discussed.
- The experimental section is relatively brief compared to the theoretical analysis. It would be valuable to see how the algorithm performs on a real-world dataset where the feature sparsity pattern is naturally occurring rather than artificially constructed.

**Questions For Authors:**

- Is there a way to use the samples from row 1 more efficiently? The current approach seems to involve significant oversampling from this row. Could this sampling strategy be optimized, and what would be the impact on the theoretical guarantees?

- How would the algorithm perform in settings with more than two clusters? Would the approach of finding discriminative features scale well, or would the sample complexity increase significantly?

- The paper focuses on the case where items within the same group have identical feature vectors. How robust is the approach to small variations within groups? Could the analysis be extended to allow for some heterogeneity is of the form that there is a linear seperator between the two groups? There is some discussion but would like to know more.

**Relation To Broader Scientific Literature:**

The paper appropriately positions its contributions within several related research areas:

- **Best arm identification literature :** The paper builds on Sequential Halving algorithms and extends these techniques to the clustering context.

- **Adaptive sensing strategies :** The three phase approach is definitely a nice novel addition to the sparse sensing literature

- **Dueling bandits :** The cleaver isolation of 1-d dueling bandit setup as a sub-sampler is a good addition.

- **Other bandit clustering problems :** Limited impact here since the setup is of a bipartite graph kind of setup.

The authors claim their approach allows for more efficient budget allocation by focusing on the most relevant features, leading to lower observation budgets than previous methods, particularly in high-dimensional settings.

**Theoretical Claims:**

I skimmed through the details of the proof, they did appear correct. My primary focus was on :

- Lemma B.1 (guarantees for CSH algorithm)

- Theorem 3.1 (upper bound)

- Theorem 4.1 (lower bound)

The theoretical analysis is generally sound. I noticed an issue in Section 3, where there appears to be a typo: $μ_{ij}-μ_{ij}$ should likely be $μ_{ic,j}-μ_{1,j}$ or similar?

---

> ### Author Rebuttal · Authors · 2025-03-31
>
> We first would like to thank you for your time and effort in reviewing it, and for your insightful questions.  We  now overview the remarks and questions that you formulate in your review:
> - **Oversampling from row 1**
>
>   First, we would like to emphasize that improving the budget compared to non-active settings requires over-sampling certain rows, which we refer to as representatives, following [Thuot et al.(2025) "Clustering with bandit feedback: breaking down the computation/information gap"]. This step is crucial for accurately estimating the means $\mu_1$ and $\mu_2$ and, ultimately, for accelerating the clustering task. This is not, per se, a weakness of our procedure but rather an essential aspect for achieving order-wise budget optimality (up to logarithmic factors). Notably, we chose row 1 as the first representative, but by symmetry, any randomly selected row could have served the same purpose.
>
>   Second, it may indeed be possible to slightly reduce the budget allocated to the first row by reusing previously sampled data.  For instance, if an index pair $(i,j)$ is identified such that $|\mu_{ij}-\mu_{1j}|$ is large, we can allocate $c\frac{n}{\hat{\Delta}^2}\log(n/\delta)$ samples to estimate $\mu_{1j}$ and $\mu_{ij}$ once and for all. These estimates could then be reused for classifying the remaining rows, reducing the confidence bounds required for perfect classification by a constant factor. However, applying such refinements in other parts of the algorithm would significantly reduce its transparency, both in terms of implementation and theoretical analysis. Moreover, the resulting improvement in the budget would be limited to a factor 2. For the final version, we will keep our method as it is, but we will clarify our motivations, and highlight potential ways to improve numerical constants.
> - **Extension to $K>2$ clusters**
>
>   In this manuscript, we focused on the case of two groups because it serves as an informative baseline for analyzing the optimal trade-off for the problem.  We will add to the appendix an algorithm that extends our method to the general case ($K>2$), together with an analysis of its budget which scales with a factor $K^2$. However, obtaining the optimal dependency with respect to $K$ is significantly more challenging. We invite you to read the detailed discussion on this point in our response to Reviewer RdCv.
> - **Robustness to heterogeneity within the clusters**
>
>   As mentioned page 8, our algorithm can be adapted to handle some degree of heterogeneity by appropriately adjusting certain tuning parameters. For instance, assume that we have prior knowledge that, for any feature $j\in[d]$, the within-group variation satisfies, $\max_{g(i)=g(i')} |\mu_{ij}-\mu_{i'j}|\leqslant c\min_{g(i) \ne g(i')} |\mu_{ij}-\mu_{i'j}|$. If $c<1/4$, our algorithm remains correct -- searching a single discriminative feature is still meaningful and enables classification. However, if the within-group heterogeneity is comparable to the inter-group differences in some features, our method could fail, and further investigation would be required. Observe also that if $c=1/2$, the problem is not identifiable anymore. We will expand on this discussion in the final version.
> - **Discussion on the optimal trade-off $H$**
>
>     We appreciate your suggestion and will provide more intuition about the complexity term $H$ (see Eq. (3)), along with a discussion on why we believe it is optimal. Also, we will base our discussion, as for the upper bound, on the simple example of sparse vectors.You can find a discussion on the intuitions about $H$ in the sparse setting in the rebuttal of review RdVc, and a discussion on its optimality for general values of $\Delta$ in the rebuttal of review z8W5.
> - **Budget comparison with the literature**
>
>   We acknowledge that further quantitative comparisons with other approaches and existing literature are missing. We will add this discussion in the final version, we will in particular discuss about the benefice of our model in high dimension settings. Thank you for pointing this out.
> - **Real-world applications**
>
>   While we described in the introduction that image labeling might be a very natural application for our algorithm, it seems challenging to process existing datasets such that we obtain data for the bandit framework. We agree that the paper would benefit from real-world experiments, but in view of the fact that this work is intended to focus on an in-depth theoretical investigation, we decided to only consider synthetical data. A possibility could be to setup experiments using crowdsourcing platforms or a group of experts. One could use image data (as done in [Herde, Marek, et al. "dopanim: A Dataset of Doppelganger Animals with Noisy Annotations from Multiple Humans." Advances in Neural Information Processing Systems 37 (2024): 51085-51117.]), but ask experts about specific features of depicted animals instead of the concrete class.

---

> > ### Comment · Reviewer_AdRS · 2025-04-04
> >
> > I would like to thank the authors for their detailed responses, especially regarding my oversampling query. I encourage the authors to incorporate these discussions (especially oversampling and budget comparision) into the final version to improve clarity and completeness of the current work.
> >
> > My concerns have been resolved.

---

### Official Review · Reviewer_z8W5 · 2025-03-16

**Overall Recommendation:** 3

**Summary:**

This paper considers the problem of classifying $ n $ items into two categories based on bandit feedback. Each item is associated with $ d $ features that vary depending on the class it belongs to. In each observation, an item and a feature are selected, and the corresponding feature value is observed with noise. Under this setting, the paper focuses on the *fixed confidence* regime, evaluating the minimum number $T$ of observations required to achieve a given accuracy level. It proposes an algorithm along with an upper bound and a nearly matching lower bound. Additionally, the proposed algorithm is evaluated through numerical experiments.

**Claims And Evidence:**

The main contributions of this paper are theoretical, and all appear to be supported by correct proofs.

**Essential References Not Discussed:**

I am not aware of any particular relevant literature not discussed.

**Experimental Designs Or Analyses:**

I was not able to spend much time verifying the correctness of the numerical experiments.

**Methods And Evaluation Criteria:**

The evaluation metric used in this paper is natural and reasonable.

**Other Comments Or Suggestions:**

There seems to be an extraneous period at the end of *Extension to a larger number of groups.* in Section 6.

**Other Strengths And Weaknesses:**

As mentioned in Section 4, one limitation is that the matching upper and lower bounds are restricted to specific cases of $ \Delta $. Additionally, the fact that the number of groups is limited to two poses a practical constraint. While Section 6 outlines a general approach for extending the method to three or more groups, obtaining tight upper and lower bounds in such cases would likely require nontrivial further analysis. That being said, this paper serves as a strong starting point for exploring broader problem settings. Overall, the paper is written in a very clear and readable manner.

**Questions For Authors:**

At the end of Section 4, it is stated, *"For more general $ \Delta $, we conjecture that the trade-off in $ H $ in (3) is optimal and unavoidable."* Could you provide any supporting facts or intuitions that lead to this conjecture?

**Relation To Broader Scientific Literature:**

The problem setting considered in this paper is a special case of the problem studied in (Ariu et al., 2024). However, to the best of my knowledge, this is the first paper to provide theoretical guarantees and analysis for this type of problem.

**Theoretical Claims:**

I have briefly checked the proofs of Theorem 3.1 and Theorem 4.1.
No particular issues were found.

---

> ### Author Rebuttal · Authors · 2025-03-31
>
> First, we would like to thank you for your time and effort in reviewing our paper. We corrected the small typo in Section 6 that you pointed out. Besides this, we would like to provide more insights regarding the two limitations you mentioned.
>
> - **Discussion on the optimal trade-off $H$ (Eq. (3)) for general $\Delta$**
>
>     In Corollary 3.2, we prove that our procedure is optimal when the gap vector is $s$-sparse with a constant magnitude $h$.  You can find a discussion on the intuitions behind $H$ on this sparse setting in the rebuttal to reviewer RdVc. For a general gap vector $\Delta$, we have good reasons to think that understanding the optimality of this trade-off for this simple example allows us to understand (at least intuitively) the optimality for general vectors.
>
>     Our upper bound is constituted of two terms. A first term $\frac{d\log(1/\delta)}{\theta||\Delta||^2}$, we proved that this  is optimal (see Thm 4.1) for the sub-problem of identifying an item in the second group. The second term is defined as this minimum  $\min_{s\in[d]} \left[\frac{d}{s\Delta_{(s)}^2} + \frac{n}{\Delta_{(s)}^2}\right] \log(1/\delta)$. Indeed, the term $\frac{n}{\Delta_{(s)}^2} \log(1/\delta)$ is the price for clustering if we use a feature with a gap $|\Delta_{(s)}|$ while $\frac{d}{s\Delta_{(s)}^2}\log(1/\delta)$ corresponds to the price for selecting a feature with a magnitude at least $|\Delta_{(s)}|$. Define the effective sparsity $s^*$ for which the minimum holds, and define the effective magnitude as $\Delta_{(s^*)}$. Intuitively, we can argue that entries significantly larger than $\Delta_{(s^*)}$ are too rare to be detected (otherwise, $s^*$ would be smaller), and entries much smaller than $\Delta_{(s^*)}$ are too weak to be used for classification. Our insight is that the problem is as hard as if the gap vector were $s^*$-sparse with a constant magnitude $\Delta_{(s^*)}$, a setting where we have matching lower and upper bounds (see corollary 3.2), leading to our complexity.
>
>     We believe that proving formally this optimality is difficult in general. We have currently a sketch of proof for which we reduce the problem into a problem of good-arm identification -- we prove that each algorithm that solves the problem should be able to identify a feature larger (up to a constant) than $\Delta_{(s^*)}$ in the gap-vector. However, we would then need an instance-dependent lower bound for such problem. Unfortunately, contrary to the problem of best-arm identification, such a lower bound does not exist yet in the literature (see [Zhao et al.(2023), Chaudhuri et Kalyanakrishnan(2019),Katz-Samuels et Jamieson(2020),  De Heide et al.(2021)]. We are currently working on such lower bound, we believe that this would be a more valuable contribution in a separate paper fully dedicated to good-arm identification.
>
>  - **Extension to $K>2$ clusters**
>
>      In this manuscript, we focused on the case of two groups because it serves as an informative baseline for analyzing the optimal trade-off in clustering with bandit feedback. This problem had not been studied before, even in the binary setting. Naturally, extending the approach to $K>2$ groups is an important and practical direction. We will add to the appendix an algorithm that extends our method to the general case where $K>2$, together with an analysis of its (non-optimal) budget. However, as you observed, obtaining the optimal dependency with respect to $K$ is significantly more challenging. We invite you to read the detailed discussion on this point in our response to Reviewer RdCv.

---

### Decision · Program_Chairs · 2025-05-01

**Decision:**

Accept (poster)

**Comment:**

This paper considers the problem of clustering items under bandit feedback in a fixed-confidence setting. It is assumed that the items are divided into two unknown groups, and items within the same group share the same $d$-dimensional feature vector. At each time step, the agent selects a pair consisting of an item and one feature (out of $d$ features), and receives noisy feedback pertaining to that feature component. The objective is to correctly cluster all items with a confidence guarantee of at least $1 - \delta$, while minimizing the number of samples used.

Previous research on fixed-confidence bandit clustering has typically assumed that the entire feature vector of an item can be observed at each time step. In contrast, this study introduces the challenge of observing only a single feature of the feature vector, leading to the difficulty of “Finding the Right Feature out of Many.” This paper is the first to provide analysis for adaptive algorithms under such a setting.

Several limitations are noted, such as the lower and upper bounds not being tight, optimality being achieved only under restricted values of $\Delta$ ((it would be good to have comparisons with the lower/upper bounds in existing studies such as Yang et al., Thuot et al., Ariu et al., and Yavas et al.), the number of clusters being limited to two, and heterogeneity within the same cluster not being considered. In terms of algorithm design, there appear to be many areas for improvement—for example, they apply an anytime version using the doubling trick from Successive Halving (SH), a method suitable for fixed-budget problems, to the fixed-confidence problem. The experiments were limited, and there was mention of a lack of ablation studies on $\theta$ and comparisons with existing methods (Ariu et al., 2024).

The authors have addressed these concerns to some extent in their rebuttal, and all reviewers seem to have agreed to assign or maintain a positive score in response. I expect these points to be reflected in the revised version.